# Don't Think Longer, Think Wisely: Optimizing Thinking Dynamics for Large Reasoning Models

**Sohyun An**[1], **Ruochen Wang**[1], **Tianyi Zhou**[2], **Cho-Jui Hsieh**[1]

[1]University of California, Los Angeles
[2]University of Maryland, College Park

sohyun0423@cs.ucla.edu    tianyi@umd.edu    chohsieh@cs.ucla.edu

## Abstract

While recent success of large reasoning models (LRMs) significantly advanced LLMs' reasoning capability by optimizing the final answer accuracy using reinforcement learning, they may also drastically increase the output length due to *overthinking*–characterized by unnecessarily complex reasoning paths that waste computation and potentially degrade the performance. We hypothesize that such inefficiencies stem from LRMs' limited capability to dynamically select the proper modular reasoning strategies, termed *thinking patterns* at the right position. To investigate this hypothesis, we propose a dynamic optimization framework that segments model-generated reasoning paths into distinct thinking patterns, systematically identifying and promoting beneficial patterns that improve the answer while removing detrimental ones. Empirical analysis confirms that our optimized thinking paths yield more concise yet sufficiently informative trajectories, enhancing reasoning efficiency by reducing attention FLOPs by up to 47% while maintaining accuracy for originally correct responses. Moreover, a non-trivial portion of originally incorrect responses are transformed into correct ones, achieving a 15.6% accuracy improvement with reduced length. Motivated by the improvement brought by the optimized thinking paths, we apply a preference optimization technique supported by a pairwise dataset contrasting suboptimal and optimal reasoning paths. Experimental evaluations across multiple mathematical reasoning benchmarks reveal that our method notably reduces computational overhead while simultaneously improving reasoning accuracy, achieving up to a 12% accuracy improvement and reducing token usage from approximately 5,000 to 3,000 tokens.

## 1 Introduction

Recent advancements in Large Reasoning Models (LRMs) [13, 44, 9] have substantially enhanced their capabilities across various complex reasoning-intensive tasks. These advancements have largely been driven by outcome-only-based reinforcement learning (RL) [23, 25, 12, 16], which train models to maximize the final answer accuracy. Models trained under this framework prioritize producing the correct answer over generating an optimal reasoning trajectory, and thus can sometimes produce long reasoning paths, *i.e.*, *test-time scaling*, achieving strong performance even in complex scenarios [28, 36, 15]. While these methods effectively increase the correctness of final predictions, they also reveal inherent limitations [19]. One notable issue frequently observed in current LRMs is the tendency toward *overthinking* [4, 29, 31], characterized by excessively prolonged reasoning paths or unnecessarily intricate inference steps. This behavior typically results in significant computational inefficiencies, increasing resource usage without improvement in performance, and can even degrade overall performance due to excessive exploration [30, 2, 27, 22].

39th Conference on Neural Information Processing Systems (NeurIPS 2025).

We hypothesize that this inefficiency largely stems from current LRMs' limited capability to dynamically identify and select optimal reasoning strategies–referred to hereafter as *thinking patterns*–at critical junctures within the reasoning process. A thinking pattern is defined as a modular reasoning segment that performs a distinct cognitive function, such as hypothesis generation, self-verification, intermediate summarization, or exploring alternative scenarios [7]. An ideal LRM should dynamically utilize beneficial thinking patterns while minimizing or discarding unnecessary or detrimental ones, thus improving computational efficiency without compromising accuracy.

To investigate this hypothesis, we formulate the enhancement of reasoning efficiency as a constrained optimization problem aimed at minimizing the expected computational cost of reasoning trajectories while preserving or improving task performance. Subsequently, we propose a framework for **D**ynamic **T**hinking pattern **O**ptimization (**DTO**), specifically designed to refine the selection of modular reasoning strategies. Our proposed method operates by first segmenting model-generated reasoning trajectories into identifiable thinking patterns. We then systematically evaluate each segment's contribution, classifying it as positive or negative based on its impact on reasoning efficiency. By identifying appropriate finalization points and selectively pruning negative segments while reinforcing positive ones, our approach not only compresses the original reasoning paths into concise, effective trajectories, but also enables the transformation of flawed reasoning traces associated with incorrect outcomes into more coherent and accurate alternatives. In contrast to prior methods, which rely heavily on heuristic truncation [4] or aggregate metrics like token length [27, 17], our framework explicitly models and optimizes the individual contributions of reasoning segments, enabling a more precise enhancement of reasoning efficiency.

To explicitly guide LRMs toward more optimal reasoning behaviors, we integrate a preference optimization approach [20], leveraging a specialized pairwise dataset that contrasts suboptimal and optimal reasoning trajectories based on our framework. This encourages models to preferentially adopt thinking patterns demonstrated to yield superior efficiency and effectiveness. We validate our framework through extensive experiments conducted across multiple established mathematical reasoning benchmarks. The results consistently demonstrate that our approach reduces computational requirements while simultaneously enhancing reasoning performance. Our contributions and findings are summarized as follows:

- We formulate the enhancement of LRM reasoning efficiency as a constrained optimization problem, and introduce a dynamic optimization framework called DTO, for this purpose.

- We empirically validate that dynamically selecting optimal thinking patterns significantly improves reasoning efficiency.

- Leveraging a preference optimization strategy based on pairwise trajectory comparisons, we achieve high reasoning efficiency across various mathematical reasoning benchmarks, attaining up to 12% higher accuracy over the original LRM, while reducing token usage from around 5,000 to 3,000.

## 2 Related Work

**Large Reasoning Models.** LLMs have traditionally exhibited rapid, intuitive decision-making aligned with *System 1* thinking. Recent efforts aim to equip them with more deliberate, *System 2* reasoning abilities. Early studies primarily adopted prompting-based methods, such as chain-of-thought, to elicit explicit step-by-step reasoning [34, 32, 39, 14]. While effective for structured reasoning, these methods often struggled with complex, multi-step problems [36]. To address this, later work introduced *test-time scaling* techniques [28, 3, 35, 24], incorporating search and verification during inference to boost accuracy. More recently, Large Reasoning Models (LRMs) have been developed to internalize sophisticated reasoning capabilities within the model itself [9, 44, 13]. They are typically trained via outcome-based RL that focuses on maximizing final answer accuracy [23, 25, 12]. While effective for improving correctness, it may inadvertently lead to *overthinking*–unnecessarily long or complex reasoning paths that reduce computational efficiency [4, 29, 31].

**Reasoning Efficiency of LRMs.** Several recent studies have addressed the issue of reasoning efficiency in LRMs. For example, some studies [22, 41, 2, 30, 1, 38, 26, 11] have introduced various length-based objectives integrated with RL approaches, while other efforts [40, 42] have employed SFT supplemented by the collection of effective datasets. Complementary to these strategies, another line of research [17, 4, 27] has investigated post-hoc refinement and scoring mechanisms for reasoning

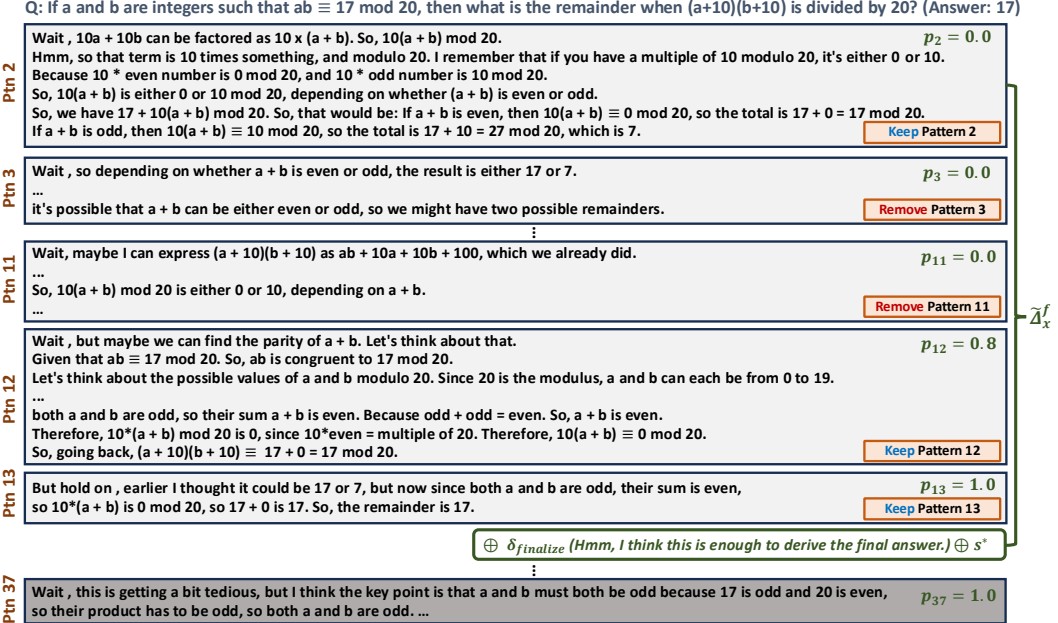

Q: If a and b are integers such that ab ≡ 17 mod 20, then what is the remainder when (a+10)(b+10) is divided by 20? (Answer: 17)

**Ptn 2** — $p_2 = 0.0$
Wait , 10a + 10b can be factored as 10 x (a + b). So, 10(a + b) mod 20.
Hmm, so that term is 10 times something, and modulo 20. I remember that if you have a multiple of 10 modulo 20, it's either 0 or 10.
Because 10 * even number is 0 mod 20, and 10 * odd number is 10 mod 20.
So, 10(a + b) is either 0 or 10 mod 20, depending on whether (a + b) is even or odd.
So, we have 17 + 10(a + b) mod 20. So, that would be: If a + b is even, then 10(a + b) ≡ 0 mod 20, so the total is 17 + 0 = 17 mod 20.
If a + b is odd, then 10(a + b) ≡ 10 mod 20, so the total is 17 + 10 = 27 mod 20, which is 7. **Keep Pattern 2**

**Ptn 3** — $p_3 = 0.0$
Wait , so depending on whether a + b is even or odd, the result is either 17 or 7.
…
it's possible that a + b can be either even or odd, so we might have two possible remainders. **Remove Pattern 3**

⋮

**Ptn 11** — $p_{11} = 0.0$
Wait, maybe I can express (a + 10)(b + 10) as ab + 10a + 10b + 100, which we already did.
…
So, 10(a + b) mod 20 is either 0 or 10, depending on a + b. **Remove Pattern 11**

$\tilde{\Delta}_x^f$

**Ptn 12** — $p_{12} = 0.8$
Wait , but maybe we can find the parity of a + b. Let's think about that.
Given that ab ≡ 17 mod 20. So, ab is congruent to 17 mod 20.
Let's think about the possible values of a and b modulo 20. Since 20 is the modulus, a and b can each be from 0 to 19.
…
both a and b are odd, so their sum a + b is even. Because odd + odd = even. So, a + b is even.
Therefore, 10*(a + b) mod 20 is 0, since 10*even = multiple of 20. Therefore, 10(a + b) ≡ 0 mod 20.
So, going back, (a + 10)(b + 10) ≡ 17 + 0 = 17 mod 20. **Keep Pattern 12**

**Ptn 13** — $p_{13} = 1.0$
But hold on , earlier I thought it could be 17 or 7, but now since both a and b are odd, their sum is even,
so 10*(a + b) is 0 mod 20, so 17 + 0 is 17. So, the remainder is 17. **Keep Pattern 13**

⊕ $\delta_{finalize}$ (Hmm, I think this is enough to derive the final answer.) ⊕ $s^*$

⋮

**Ptn 37** — $p_{37} = 1.0$
Wait , this is getting a bit tedious, but I think the key point is that a and b must both be odd because 17 is odd and 20 is even,
so their product has to be odd, so both a and b are odd. …

Figure 1: **Illustration of DTO.** We construct a truncated reasoning trajectory $\Delta_x^f$ by identifying the point where the probability score $p_i$ in Equation (6) exceeds a threshold $T = 1.0$, and then applying the binary selection function $f(\cdot)$ from Equation (7). We then append the finalization pattern $\delta_{\text{finalize}}$ and sampled answer $s^*$ (Equation (9)) to form $\tilde{\Delta}_x^f$. Finally, the pruning function $g(\cdot)$ (Equation (11)) refines the trajectory into the optimized version $\Delta_x^g$, as illustrated in the orange box.

trajectories generated by models, demonstrating promising results with relatively low computational overhead. Our method builds on this line of work, leveraging its demonstrated efficiency and broad applicability. Although primarily evaluated on base LRMs, these approaches can be generally applied to models trained or fine-tuned through other efficiency-enhancing strategies. More specifically, Luo et al. [17] introduced the Length-Harmonizing score, combining reasoning accuracy and token length within a PPO-style fine-tuning objective to encourage more concise reasoning paths. Preference-based optimization approaches, as explored by Chen et al. [4] and Shen et al. [27], have also proven effective. These methods construct pairwise datasets to guide models toward more efficient reasoning. Specifically, Chen et al. [4] generates positive samples by instructing an LLM to truncate correct reasoning at the point where an initial solution is followed by a reflective step, while Shen et al. [27] employs a scoring mechanism favoring shorter correct and longer incorrect responses when forming pairwise comparisons. Nevertheless, these methods primarily rely on aggregate-level metrics such as token length or heuristic truncation points. As a result, they are limited in their ability to precisely assess and optimize the contribution of individual reasoning segments, potentially hindering more optimal efficiency. To address this limitation, we propose a dynamic optimization framework that operates at the segment level.

## 3 Dynamic Optimization of Thinking Patterns

Reasoning trajectories generated by LRMs exhibit a discernible structure, where distinct segments serve specific cognitive roles such as self-verification, intermediate summarization, or exploring alternatives [7]. We refer to these segments as *thinking patterns*, which can often be identified through linguistic cues such as *"Wait"*, *"Alternatively"* that signal shifts in reasoning strategy. As illustrated in Figure 1, certain thinking patterns promote reasoning progress by facilitating equation formulation and step-by-step computation. Conversely, other patterns hinder efficiency by repeating prior reasoning or continuing unnecessary verification without contributing to the reasoning process. We hypothesize that such inefficiencies in LRMs stem from their limited ability to dynamically select appropriate thinking patterns at critical moments. For example, models may fail to terminate reasoning even after sufficient evidence has been gathered. To investigate this hypothesis, we formulate the enhancement of reasoning efficiency as a constrained optimization problem, aiming

to reduce computational cost while enforcing a lower bound on task performance. Building on this formulation, we introduce a dynamic optimization framework termed **DTO** that identifies and prunes unproductive segments and reinforces those that contribute positively.

## 3.1 Problem Formulation

Let $\Delta_x$ be a finite list of thinking patterns for problem $x \sim \mathcal{D}$, *i.e.*, $\Delta_x = [\delta_1, \delta_2, \ldots, \delta_{n_x}]$, where $\mathcal{D}$ denotes the underlying distribution over input problems. Each thinking pattern $\delta$ incurs a non-negative computational cost $c(\delta) \geq 0$, typically quantified by metrics such as token length or FLOPs. The total cost of the reasoning trajectory $\Delta_x$ is therefore given by:

$$\mathcal{C}(\Delta_x) = \sum_{\delta \in \Delta_x} c(\delta). \tag{1}$$

Here, our objective is to construct reasoning trajectories that integrate effective thinking patterns to achieve high reasoning efficiency. Formally, we define the constrained optimization problem as follows, aiming to minimize the expected computational cost while maintaining task performance at or above its prior level:

$$\begin{aligned} \underset{\Delta_x = [\delta_1, \ldots, \delta_{n_x}]}{\text{minimize}} \quad & \mathbb{E}_{x \sim \mathcal{D}}[\mathcal{C}(\Delta_x)] \\ \text{subject to} \quad & \mathbb{E}_{x \sim \mathcal{D}}[\mathcal{P}(\Delta_x)] \geq \alpha, \end{aligned} \tag{2}$$

where $\mathcal{P}(\cdot)$ denotes a task-specific performance metric such as final answer accuracy, and $\alpha$ represents a lower bound on the expected task performance, *e.g.,* the expected performance of the original trajectories generated by a base LRM if we aim to maintain the original performance.

## 3.2 DTO: Constructing Optimal Reasoning Trajectories

To solve the problem defined in Equation (2), we must first characterize what constitutes an optimal trajectory under this formulation. We define such a trajectory as one that incrementally generates thinking patterns conducive to achieving the reasoning objective, such as narrowing down possibilities or providing key insights, and terminates the reasoning process once sufficient information has been accumulated to produce a correct answer. To extract such optimal reasoning trajectory from an LRM-generated response, we follow a two-step procedure: (1) identify the appropriate point at which the reasoning should be finalized, and (2) prune intermediate thinking patterns that do not meaningfully contribute to the reasoning objective.

Given a reasoning trajectory $y$ generated by a LRM $\pi_\theta$ for a problem $x$ with a known ground-truth answer $a^*$, we segment $y$ into a sequence of distinct thinking patterns:

$$y = [\delta_1, \ldots, \delta_{n_y}], \quad y \sim \pi_\theta(x). \tag{3}$$

To systematically determine optimal termination points, inspired by [6], we define an *exit* thinking pattern, $\delta_{\text{exit}}$, expressed as "`... Wait, I suddenly got the final answer to the whole problem. Final Answer: \boxed{`". For each candidate termination point $i$, we construct a partial trajectory $\tau_i$ by appending the exit pattern:

$$\tau_i = \delta_1 \oplus \delta_2 \oplus \cdots \oplus \delta_i \oplus \delta_{\text{exit}}, \quad \text{for } i = 1, \ldots, n_y. \tag{4}$$

We then employ Monte Carlo estimation, sampling multiple completions:

$$\mathcal{R}_i = \{r_1, r_2, \ldots, r_M\}, \quad r_j \sim \pi_\theta(\tau_i). \tag{5}$$

Benefiting from highly efficient and parallelizable inference frameworks [45], and by restricting generation to a small number of tokens with a slight margin over the token length of the ground-truth answer $a^*$, this process can be executed in a lightweight manner. We then determine the probability $p_i$ of deriving the correct answer at index $i$ by:

$$p_i = |\{r \in \mathcal{R}_i \mid a^* \in r\}|/|\mathcal{R}_i|. \tag{6}$$

We define the earliest index $i = i'$ such that $p_i$ exceeds a predefined threshold $T$, and apply a binary selection function over the thinking patterns:

$$f(\delta_i) = \begin{cases} 1 & \text{if } \delta_i \leq i' \\ 0 & \text{otherwise,} \end{cases} \tag{7}$$

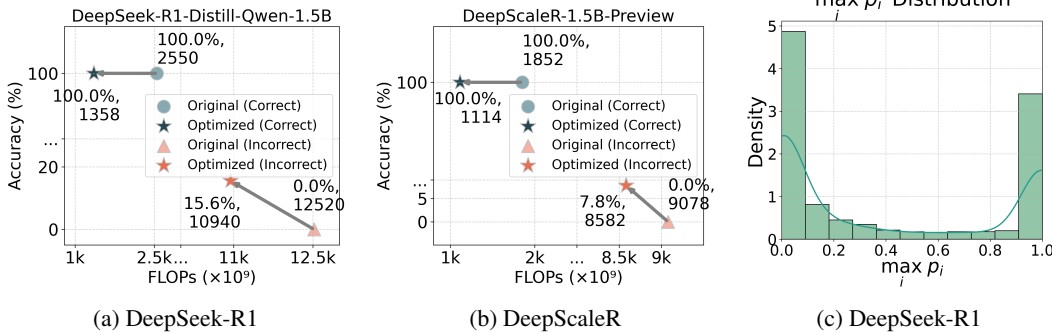

(a) DeepSeek-R1  (b) DeepScaleR  (c) DeepSeek-R1

Figure 2: **Comparison of dynamically optimized vs. original responses, and $\max_i p_i$ distributions in incorrect cases. (a), (b)** Dynamic optimization preserves accuracy for correct responses while reducing attention FLOPs (47%, 40%), and improves accuracy for incorrect ones (15.6%, 7.8%) with lower FLOPs. **(c)** shows $\max_i p_i$, the maximum estimated correctness probability across thinking patterns (Equation (6)). High values suggest that even incorrect trajectories often contain a promising intermediate segment.

yielding the truncated reasoning trajectory:

$$\Delta_x^f = [\delta_i \in y \mid f(\delta_i) = 1], \tag{8}$$

where $y$ is the whole trajectory in Equation (3).

Next, we introduce a special *finalize* thinking pattern, $\delta_{\text{finalize}}$, which signals that sufficient reasoning has been conducted—similar to how a human might intuitively recognize when to stop deliberating and provide an answer. This is expressed as "`Hmm, I think this is enough to derive the final answer.`" Unlike $\delta_{\text{exit}}$, $\delta_{\text{finalize}}$ coherently concludes the reasoning process. We sample completions conditioned on the finalized trajectory:

$$\mathcal{S}_{\Delta_x^f} = \{s_1, s_2, \ldots, s_K\}, \quad s_j \sim \pi_\theta(\Delta_x^f \oplus \delta_{\text{finalize}}). \tag{9}$$

Among the correct completions in $\mathcal{S}_{\Delta_x^f}$, we select the shortest one, $s^*$, and construct a refined reasoning trajectory:

$$\tilde{\Delta}_x^f = \Delta_x^f \oplus \delta_{\text{finalize}} \oplus s^*. \tag{10}$$

To further optimize $\tilde{\Delta}_x^f$, we evaluate the utility of each intermediate thinking pattern using an auxiliary LLM $\mu_\phi$. Given a dedicated prompt (see Appendix A.4), the model determines whether each $\delta_i \in \tilde{\Delta}_x^f$ meaningfully contributes to deriving $a^*$. Note that we instructed the model to remove segments only if they add no meaningful content, are factually incorrect in a harmful way, or are entirely off-topic or unhelpfully redundant, so as to avoid compromising the model's reasoning capability. To further ensure that the pruned sequence, which excludes $\delta_i$, still leads to the correct answer, we consider the following trajectory:

$$\tilde{\Delta}_x^{f \setminus \delta_i} = \left[ \delta_j \in \tilde{\Delta}_x^f \mid \delta_j \neq \delta_i, \text{ and before "\boxed"} \right],$$

and perform a quick decoding step for validation. Patterns that are deemed redundant and pass validation are removed via the following pruning function:

$$g(\delta_i) = \begin{cases} 1 & \text{if } \delta_i \text{ can be removed from } \tilde{\Delta}_x^f \\ 0 & \text{otherwise.} \end{cases} \tag{11}$$

This results in the final optimized reasoning trajectory as follows:

$$\Delta_x^g = [\delta_j \in \tilde{\Delta}_x^f \mid g(\delta_j) \neq 1]. \tag{12}$$

Please refer to Appendix A.3 for the full algorithm of our framework.

### 3.3 Analysis

In this section, we compare the original trajectory $y$ sampled from the base LRM with the optimized trajectory $\Delta_x^g$ defined in Equation (12).

**Setup.** We use a dataset consisting of 5,000 samples drawn from the MATH [10] training set[1] and employ the DeepSeek-R1-Distill-Qwen-1.5B [9] and DeepScaleR-1.5B-Preview [18] as $\pi_\theta$ and Llama-3.3-70B-Instruct [8] as an auxiliary LLM $\mu_\phi$. For each problem, we sample 4 responses. If at least one response is correct, we construct a reasoning trajectory $\Delta_x^g$ by applying our proposed framework to a randomly selected correct response. If no correct responses are available, we instead apply the same procedure to a randomly selected incorrect response. We set the temperature to 0.6, top_p to 0.95, and max_tokens to 8192. The value of $M$ in Equation (5) was set to 10, $K$ in Equation (9) was set to 4, and the threshold $T$ is set to 1.0, signaling that the model has accumulated sufficient information to produce the correct answer.

**Results.** As illustrated in Figure 2a and Figure 2b, the responses optimized using our method require substantially fewer attention FLOPs compared to the original responses from the base LRM. Specifically, for responses that originally produced a correct answer, **our method reduces attention FLOPs consumption by 47% and 40% for DeepSeek-R1-Distill-Qwen-1.5B and DeepScaleR-1.5B-Preview, respectively, while preserving correctness.** Interestingly, for responses that initially yielded incorrect outcomes, applying our framework to induce more effective thinking patterns such as finalizing reasoning at appropriate termination points **transforms a non-trivial portion of them into correct ones, improving accuracy by 15.6% and 7.8% for DeepSeek-R1-Distill-Qwen-1.5B and DeepScaleR-1.5B-Preview, respectively, while also reducing FLOPs consumption.** To better understand why this is possible, we analyze $\max_i p_i$ for responses that were originally incorrect in Figure 2c and Figure 4. As described in Equation (6), $p_i$ denotes a Monte Carlo estimate of the probability that the model produces the correct answer when terminated at the $i$-th thinking pattern. We find that, even in incorrect responses, many intermediate segments exhibit high $p_i$ values, suggesting that the model temporarily arrives at high-quality partial reasoning steps. This observation indicates that our method can effectively restructure these responses with more appropriate thinking patterns, thereby improving accuracy while consistently lowering the computational cost.

## 4 Preference Optimization Towards Optimal Reasoning Behaviors

Thus far, we have demonstrated that our dynamic optimization framework enables the construction of significantly more efficient reasoning trajectories by selecting appropriate thinking patterns at the right points in the reasoning process. In this section, we leverage the framework to construct a contrasting pair dataset of optimized and suboptimal reasoning trajectories, and examine the impact of applying a preference optimization technique using this dataset.

### 4.1 Preference Optimization with Dynamically Optimized Trajectories

To apply the preference optimization technique, we first construct a pairwise dataset $(y_w, y_l)$ for each problem $x \sim \mathcal{D}$. For each $x$, we aim to generate $N$ response pairs. Specifically, we begin by sampling 4 responses from the LRM $\pi_\theta$. Let $N_c$ denote the number of correct responses among them. If $N_c \geq N$, we select the $N$ shortest correct responses and apply our dynamic optimization framework to generate optimized trajectories, which are used as $y_w$. If $N_c < N$, we select all $N_c$ correct responses and additionally sample $N - N_c$ incorrect responses at random. We then apply our framework to these $N$ responses to obtain optimized trajectories as $y_w$. For each $y_w$, we select the longest unoptimized trajectory among the sampled responses to serve as the corresponding $y_l$, thereby constructing the final pairwise dataset $\mathcal{D}'$. We then apply SimPO [20], which demonstrates strong performance while incurring relatively low computational cost, in order to guide the LRM toward more optimal reasoning behaviors:

$$\mathcal{L}_{\text{SimPO}}(\pi_\theta) = -\mathbb{E}_{(x,y_w,y_l)\sim\mathcal{D}'}\left[\log\sigma\left(\frac{\beta}{|y_w|}\log\pi_\theta(y_w \mid x) - \frac{\beta}{|y_l|}\log\pi_\theta(y_l \mid x) - \gamma\right)\right], \quad (13)$$

where $\beta$ and $\gamma$ are hyperparameters, $\sigma(\cdot)$ denotes the sigmoid function, and $|y_w|$ and $|y_l|$ represent the token lengths of $y_w$ and $y_l$, respectively. Please refer to Appendix A for more details.

---

[1]We followed the split used in Luo et al. [17]

Table 1: **Comparison with other methods on the DeepSeek-R1-Distill-Qwen-1.5B model.** We evaluate the effectiveness of our method using a dynamic optimization framework and a preference optimization technique applied to DeepSeek-R1-Distill-Qwen-1.5B by comparing it against existing methods.

| Method | MATH | | | GSM8K | | | Gaokao | | |
|---|---|---|---|---|---|---|---|---|---|
| | Acc. (↑) | #Tokens (↓) | Eff. (↑) | Acc. (↑) | #Tokens (↓) | Eff. (↑) | Acc. (↑) | #Tokens (↓) | Eff. (↑) |
| Instruct ver. [37] | 76.36 | 555.16 | N/A | 85.37 | 315.44 | N/A | 65.13 | 575.86 | N/A |
| Baseline | 79.80 | 3543.44 | 1.000 | 82.13 | 1382.99 | 1.000 | 66.62 | 3725.16 | 1.000 |
| Fast Prompt | 81.17 | 3354.99 | 1.074 | 85.14 | 1894.73 | 0.757 | 69.68 | 3634.30 | 1.072 |
| SFT | 81.28 | 3180.10 | 1.135 | 80.12 | 933.89 | 1.445 | 67.34 | 3245.37 | 1.160 |
| O1-Pruner [17] | 82.31 | 2593.06 | 1.409 | 80.67 | 669.41 | **2.029** | 66.69 | 2827.81 | 1.319 |
| DAST [27] | 83.35 | 2817.94 | 1.313 | 84.02 | 1174.89 | 1.204 | 69.42 | 3058.96 | 1.269 |
| FCS + Ref. [4] | 84.72 | 2548.55 | 1.476 | 84.29 | 1080.19 | 1.314 | 71.30 | 2750.35 | 1.450 |
| DTO (Ours) | 85.48 | 1936.19 | **1.960** | 83.91 | 844.18 | 1.674 | 72.66 | 2137.59 | **1.901** |

| Method | AMC2023 | | | AIME2024 | | | AIME2025 | | |
|---|---|---|---|---|---|---|---|---|---|
| | Acc. (↑) | #Tokens (↓) | Eff. (↑) | Acc. (↑) | #Tokens (↓) | Eff. (↑) | Acc. (↑) | #Tokens (↓) | Eff. (↑) |
| Instruct ver. [37] | 54.75 | 786.43 | N/A | 11.22 | 956.11 | N/A | 8.11 | 887.40 | N/A |
| Baseline | 58.25 | 5338.54 | 1.000 | 21.44 | 7359.24 | 1.000 | 18.89 | 7236.66 | 1.000 |
| Fast Prompt | 61.00 | 5073.15 | 1.102 | 21.56 | 7261.19 | 1.019 | 20.33 | 7137.90 | 1.091 |
| SFT | 61.08 | 5030.76 | 1.113 | 23.78 | 7151.84 | 1.141 | 18.44 | 7122.97 | 0.992 |
| O1-Pruner [17] | 65.50 | 4370.83 | 1.373 | 21.78 | 7015.30 | 1.066 | 17.67 | 6742.34 | 1.004 |
| DAST [27] | 66.58 | 4590.91 | 1.329 | 24.00 | 7077.45 | 1.164 | 19.78 | 6846.85 | 1.107 |
| FCS + Ref. [4] | 68.92 | 4166.60 | 1.516 | 24.44 | 6698.77 | 1.252 | 20.67 | 6545.62 | 1.210 |
| DTO (Ours) | 70.25 | 3376.98 | **1.907** | 28.00 | 5877.44 | **1.635** | 21.11 | 5689.38 | **1.421** |

## 4.2 Experiments

**Dataset.** We evaluate our method against various existing approaches across several established mathematical reasoning benchmarks, including the test sets of MATH [10], GSM8K [5], Gaokao (Mathematics) [43], AMC2023, AIME2024, and AIME2025.

**Baselines.** In line with prior works [4, 27, 17], we compare our method against the following baselines: **Instruct ver.** refers to the instruction-tuned version of the base model on which the LRM is trained. **Baseline** denotes the base LRM $\pi_\theta$ without any additional prompting or fine-tuning. **Fast Prompt** [17] appends the instruction *"Solve the problem as quickly as possible"* to the original prompt (see Appendix A.4) and uses the resulting prompt to generate responses. **SFT** represents supervised fine-tuning of $\pi_\theta$ using $N$ correct and shortest responses from the model. **O1-Pruner** [17] scores reasoning trajectories generated by the LRM using the Length-Harmonizing score, which jointly considers accuracy and token length. $N$ trajectories per problem $x$ are then used to fine-tune the model via a PPO-style objective. **DAST** [27] introduces a self-defined token length budget, formulated as a linear combination of the average token length of correct responses and a predefined maximum generation length. Leveraging this, it assigns preference scores to responses sampled from a LRM, favoring shorter correct responses and longer incorrect ones during pairwise data construction. A response pair is formed when the score margin exceeds 0.3, and the resulting dataset is used to apply a preference optimization method with at most $N$ trajectories per problem $x$. **FCS + Ref.** [4] constructs a pairwise dataset by leveraging heuristic truncation points in LRM-generated trajectories. Specifically, when a correct response is present, the LLM $\mu_\phi$ is prompted to identify the locations of the First Correct Solution and the subsequent Reflection. The segment up to the reflection point is extracted as the positive example (at most $N$ per problem $x$), while the longest trajectory among the sampled responses is selected as the negative example. Preference optimization technique is then applied to problems for which such pairwise data can be constructed—that is, when at least one correct response is available.

**Metrics.** For each dataset, we report both the average accuracy and the average number of generated tokens. Additionally, inspired by Qu et al. [21], we compute the following efficiency metric $\eta$:

$$\eta = \frac{\mathbb{E}_{x\sim\mathcal{D},\, y\sim\pi_{\theta^*}(x)}[\mathcal{P}(y)]}{\mathbb{E}_{x\sim\mathcal{D},\, y_0\sim\pi_\theta(x)}[\mathcal{P}(y_0)]} \cdot \frac{\mathbb{E}_{x\sim\mathcal{D},\, y_0\sim\pi_\theta(x)}[\mathcal{C}(y_0)]}{\mathbb{E}_{x\sim\mathcal{D},\, y\sim\pi_{\theta^*}(x)}[\mathcal{C}(y)]} \tag{14}$$

Here, $\mathcal{P}(\cdot)$ measures final answer accuracy, $\mathcal{C}(\cdot)$ denotes the number of generated tokens, $\pi_\theta$ represents the original base model, while $\pi_{\theta^*}$ corresponds to the model after applying a specific method. A higher value of $\eta$ indicates a more favorable trade-off between inference efficiency and performance.

Table 2: **Comparison with other methods on the DeepScaleR-1.5B-Preview.** We evaluate the effectiveness of our method using a dynamic optimization framework and a preference optimization technique applied to DeepScaleR-1.5B-Preview by comparing it against existing methods.

| | MATH | | | GSM8K | | | Gaokao | | |
| Method | Acc. (↑) | #Tokens (↓) | Eff. (↑) | Acc. (↑) | #Tokens (↓) | Eff. (↑) | Acc. (↑) | #Tokens (↓) | Eff. (↑) |
|---|---|---|---|---|---|---|---|---|---|
| Instruct ver. [37] | 76.36 | 555.16 | N/A | 85.37 | 315.44 | N/A | 65.13 | 575.86 | N/A |
| Baseline | 88.48 | 2700.37 | 1.000 | 87.38 | 1601.98 | 1.000 | 75.97 | 3053.67 | 1.000 |
| Fast Prompt | 89.41 | 2601.49 | 1.049 | 88.38 | 1652.94 | 0.980 | 77.01 | 2941.73 | 1.052 |
| SFT | 89.19 | 2634.80 | 1.033 | 86.43 | 1469.03 | 1.079 | 76.23 | 2950.22 | 1.039 |
| O1-Pruner [17] | 89.74 | 2259.56 | 1.212 | 86.41 | 1235.48 | 1.282 | 77.47 | 2539.68 | 1.226 |
| DAST [27] | 89.74 | 2551.49 | 1.073 | 86.71 | 1502.93 | 1.058 | 77.21 | 2880.06 | 1.078 |
| FCS + Ref. [4] | 88.81 | 2247.38 | 1.206 | 88.10 | 1254.06 | 1.288 | 77.21 | 2531.05 | 1.226 |
| DTO (Ours) | 89.14 | 1994.27 | **1.364** | 87.23 | 1184.53 | **1.350** | 77.01 | 2269.98 | **1.364** |

| | AMC2023 | | | AIME2024 | | | AIME2025 | | |
| Method | Acc. (↑) | #Tokens (↓) | Eff. (↑) | Acc. (↑) | #Tokens (↓) | Eff. (↑) | Acc. (↑) | #Tokens (↓) | Eff. (↑) |
|---|---|---|---|---|---|---|---|---|---|
| Instruct ver. [37] | 54.75 | 786.43 | N/A | 11.22 | 956.11 | N/A | 8.11 | 887.40 | N/A |
| Baseline | 76.88 | 4268.77 | 1.000 | 36.67 | 6455.16 | 1.000 | 29.17 | 6420.07 | 1.000 |
| Fast Prompt | 76.25 | 4119.06 | 1.028 | 30.83 | 6505.43 | 0.834 | 27.50 | 6338.55 | 0.955 |
| SFT | 71.88 | 4405.87 | 0.906 | 35.83 | 6755.42 | 0.934 | 27.50 | 6192.65 | 0.977 |
| O1-Pruner [17] | 79.38 | 3831.73 | 1.150 | 35.83 | 6242.41 | 1.010 | 28.33 | 6086.82 | 1.024 |
| DAST [27] | 78.75 | 4158.48 | 1.051 | 32.50 | 6551.17 | 0.873 | 27.50 | 6462.76 | 0.937 |
| FCS + Ref. [4] | 77.50 | 3755.56 | 1.146 | 35.00 | 6219.26 | 0.991 | 26.67 | 6016.55 | 0.976 |
| DTO (Ours) | 77.50 | 3403.56 | **1.264** | 38.33 | 5742.61 | **1.175** | 27.50 | 5820.90 | **1.040** |

**Results.** Here, we set $N = 2$, with the remaining experimental setup following Section 3.3. Table 1 and Table 2 summarize the performance comparisons between our method and existing approaches on two models: DeepSeek-R1-Distill-Qwen-1.5B and DeepScaleR-1.5B-Preview, evaluated across six benchmark datasets. Overall, **our approach achieves superior efficiency metrics on almost all datasets and models.** Specifically, on DeepSeek-R1-Distill-Qwen-1.5B (Table 1), DTO significantly outperforms existing methods in efficiency while maintaining comparable or better accuracy on 5 out of 6 datasets. For challenging benchmarks such as AMC and AIME, DTO improves average accuracy by almost 7% over the base model, while also reducing token usage by approximately 1,700 tokens. This improvement likely stems from DTO's ability to transform some initially incorrect trajectories into correct reasoning paths, even in cases where no correct sampled trajectory is available, which often indicates that the problem is particularly difficult. On the other hand, in such cases, FCS+Ref. fails to construct a pairwise dataset, while DAST relies solely on token length, *i.e.,* it assigns higher scores to longer incorrect responses, which may limit its ability to generalize. On DeepScaleR-1.5B-Preview (Table 2), DTO continues to achieve the highest efficiency, though with narrower margins due to this model's inherently higher accuracy and lower baseline token counts. Nonetheless, DTO consistently delivers a favorable balance between accuracy and computational efficiency.

### 4.3 Analysis

In this section, we present several analyses to validate the effectiveness of our framework using the DeepSeek-R1-Distill-Qwen-1.5B model.

**Comparison with Token-Limited Baselines.** In addition to Section 4, we further compared against a strong baseline assessing the impact of varying token limits on the base model's performance. This baseline involves forcing the base model (DeepSeek-R1-Distill-Qwen-1.5B) to produce a final answer once it reaches a maximum token limit (1024, 2048, 4096, and 6144). To facilitate this forced finalization, we appended the instruction: "`Wait, I suddenly got the final answer to the whole problem. \n\nFinal Answer:`". As shown in Table 3, the results exhibit a consistent trend: as the token limit increases, both accuracy and the number of generated tokens increase, indicating that longer sequences enable more complete reasoning. However, our method, DTO, achieves higher accuracy while using fewer or comparable tokens, highlighting its strength in both reasoning quality and efficiency.

**Frequency of Thinking Pattern Transitions.** We further examine the frequency of thinking pattern transitions by analyzing the occurrence of the linguistic cue *"Wait"*. As shown in Figure 3,

Table 3: **Comparison against the base model with varying token limits.** DTO (Ours) consistently achieves higher accuracy while using fewer or comparable tokens.

| Dataset | Base + 1024 | | Base + 2048 | | Base + 4096 | | Base + 6144 | | DTO (Ours) | |
|---|---|---|---|---|---|---|---|---|---|---|
| | Acc.(↑) | #Toks(↓) | Acc.(↑) | #Toks(↓) | Acc.(↑) | #Toks(↓) | Acc.(↑) | #Toks(↓) | Acc.(↑) | #Toks(↓) |
| MATH | 63.59 | 1009.65 | 74.78 | 1783.91 | 78.55 | 2642.43 | 79.97 | 3124.27 | 85.48 | 1936.19 |
| GSM8K | 79.40 | 576.62 | 80.48 | 722.42 | 79.78 | 850.67 | 80.12 | 928.66 | 83.91 | 844.18 |
| Gaokao | 50.06 | 957.04 | 57.86 | 1661.44 | 65.26 | 2551.03 | 67.86 | 3092.08 | 72.66 | 2137.59 |
| AMC2023 | 32.50 | 1022.44 | 50.00 | 1965.05 | 67.50 | 3416.82 | 68.12 | 4385.90 | 70.25 | 3376.98 |
| AIME2024 | 3.33 | 1024.00 | 14.17 | 2048.00 | 20.83 | 4009.87 | 24.17 | 5686.45 | 28.00 | 5877.44 |
| AIME2025 | 1.67 | 1024.00 | 12.50 | 2048.00 | 17.50 | 3986.50 | 18.33 | 5715.72 | 21.11 | 5689.38 |

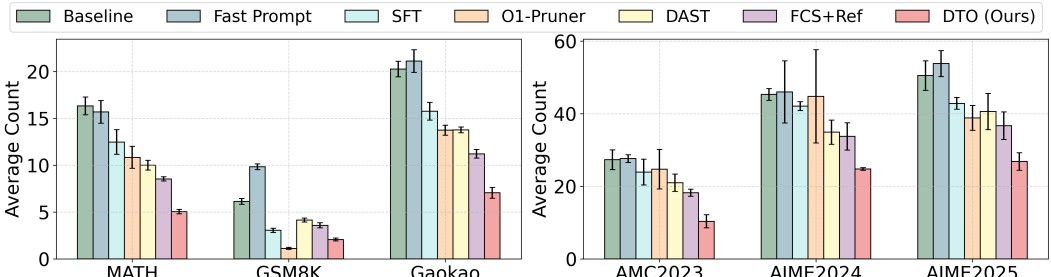

Figure 3: **The average count of *"Wait"*, which is one of the words signaling a thinking pattern transition.** Compared to all baselines, our framework generally results in the lowest average count of *"Wait"*, suggesting more concise and less interrupted reasoning trajectories. The results are averaged over 4 runs.

all methods exhibit increased frequencies as task difficulty rises. Notably, DTO consistently achieves the lowest average frequency of *"Wait"* across multiple benchmarks, while maintaining comparable or superior accuracy relative to all baseline methods, with the exception of GSM8k (see Table 1). This result suggests that **DTO effectively reduces unnecessary cognitive shifts during reasoning, leading to more concise and coherent reasoning trajectories.**

**Qualitative Analysis of Generated Reasoning Trajectories.** As shown in Table 4, our optimized reasoning trajectory achieves the same correct answer with significantly fewer tokens (814 vs. 1790) compared to the baseline. The baseline response exhibits signs of overthinking, including repetitive verification, redundant restatements, and unnecessary self-correction. In contrast, our method generates a more concise trajectory by directly applying the relevant formula and confidently progressing through the necessary steps. This example illustrates how **dynamic reasoning optimization effectively reduces token overhead without compromising correctness, producing more efficient reasoning paths.**

**Generalization to different domains.** We evaluate the generalization capability of our framework by examining whether a model trained on the MATH dataset, as detailed in the previous section, can improve reasoning efficiency beyond the mathematical domain. To this end, we use the MMLU-Pro dataset [33], which comprises challenging, reasoning-centric questions across a wide range of domains. Among these, we conduct evaluations on three domains using 100 randomly sampled questions per domain and the same five CoT few-shot examples across all methods. As shown in Table 5, our method consistently achieves the highest reasoning efficiency across all three domains, significantly reducing token usage while maintaining comparable accuracy to baselines. This suggests that **our framework generalizes well beyond the mathematical domain, effectively condensing reasoning in tasks that differ from the training domain**.

## 5 Conclusion

In this paper, we introduced a dynamic optimization framework termed DTO to mitigate the inefficiencies arising from *overthinking* in LRMs. By dynamically identifying and optimizing modular reasoning strategies, our approach substantially reduces computational overhead while improving reasoning accuracy. Empirical evaluations across diverse benchmarks demonstrate the effective-

Table 4: **Comparison of Reasoning Trajectories between Baseline and DTO.** Due to space constraints, the middle part has been replaced with "...". The segment highlighted in yellow redundantly restates the same fact as the one in green. For more details, please refer to Appendix B.

**Question**: "What is the degree measure of the acute angle formed by lines with slopes 2 and $\frac{1}{3}$?" (Answer: 45)

**Baseline (1790 tokens):** "Okay, so I need to find the acute angle between two lines with slopes 2 and 1/3. ... So, if the slopes are $m_1$ and $m_2$, the angle $\theta$ between them is given by: $\theta = \arctan(|m_2 - m_1|/(1 + m_1 m_2))$. Is that right? Let me check. Yeah, I think that's correct because the tangent of the angle between two lines is equal to the absolute value of $(m_2 - m_1)/(1 + m_1 m_2)$. ... So, putting it together, we have: $\theta = \arctan(|(-5/3)/(5/3)|) = \arctan(|-1|) = \arctan(1)$. Because the absolute value of -1 is 1. So, $\arctan(1)$ is... $45°$, right? Since $\tan(45°) = 1$. Wait, but hold on, the question asks for the acute angle. So, $45°$ is already acute, since it's less than $90°$. So, is that the answer? It seems straightforward. But let me make sure ... Slope $m_1 = 2, m_2 = 1/3$. ... So, $\arctan(1)$ is $45°$. That seems correct. Wait a second, but I remember that the angle between two lines can also be found using the formula: $\tan(\theta) = |(m_2 - m_1)/(1 + m_1 m_2)|$. ... **Final Answer:** The acute angle formed by the lines is $\boxed{45}$."

**DTO (814 tokens):** "Alright, so I need to find the acute angle formed by two lines with slopes 2 and 1/3. ... Let me write that down: $\tan(\theta) = |(m_2 - m_1)|/|1 + m_1 m_2|$. So substituting the values: $\tan(\theta) = |(1/3 - 2)/(1 + (2) * (1/3))|$. ... Oh, so $\tan(\theta)$ is 1. Then $\theta$ is arctan of 1. I remember that $\tan(45°)$ is 1, so $\theta$ is $45°$. Since we're asked for the acute angle, $45°$ is already acute because it's less than $90°$. So, I think that's it. The acute angle is 45 °. **Final Answer:** The acute angle is $\boxed{45}$."

Table 5: **Evaluation on MMLU-Pro to validate generalization beyond mathematics.** All methods use the same five CoT few-shot examples.

| Method | Law | | | Chemistry | | | Engineering | | |
| --- | --- | --- | --- | --- | --- | --- | --- | --- | --- |
| | Acc. (↑) | #Tokens (↓) | Eff. (↑) | Acc. (↑) | #Tokens (↓) | Eff. (↑) | Acc. (↑) | #Tokens (↓) | Eff. (↑) |
| Instruct ver. [37] | 11.00 | 183.98 | N/A | 21.00 | 226.18 | N/A | 20.25 | 377.11 | N/A |
| Baseline | 17.50 | 1716.77 | 1.000 | 40.75 | 4427.44 | 1.000 | 22.00 | 6173.27 | 1.000 |
| Fast Prompt | 17.00 | 1685.09 | 0.990 | 37.25 | 4508.80 | 0.898 | 20.75 | 5972.20 | 0.975 |
| SFT | 17.75 | 1590.35 | 1.095 | 39.25 | 4270.96 | 0.998 | 19.25 | 6038.40 | 0.895 |
| O1-Pruner [17] | 16.25 | 995.03 | 1.602 | 39.25 | 3688.95 | 1.156 | 23.00 | 5225.53 | 1.235 |
| DAST [27] | 16.25 | 1180.12 | 1.351 | 38.00 | 4433.26 | 0.931 | 23.75 | 6137.51 | 1.086 |
| FCS + Ref. [4] | 15.75 | 1034.84 | 1.493 | 39.25 | 4096.80 | 1.041 | 22.75 | 5875.08 | 1.087 |
| DTO (Ours) | 16.75 | 734.20 | **2.238** | 39.75 | 3142.64 | **1.374** | 23.00 | 4186.74 | **1.542** |

ness of our method when combined with preference optimization, underscoring the importance of strategically managing reasoning processes in LRMs.

# Acknowledgements

This work is partially supported by NSF 2048280, 2325121, 2244760, 2331966 and ONR N00014-23-1-2300:P00001.

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

## A    Experimental Details

### A.1    The Distribution of Maximum Correctness Probability for Incorrect Responses.

Due to space limitations, we report the distribution of $\max_i p_i$ for incorrect responses of the DeepScaleR-1.5B-Preview model in Figure 4. In Section 3.3, we analyze the distribution of $\max_i p_i$ values computed from reasoning trajectories that initially resulted in incorrect answers. Here, $p_i$ denotes a Monte Carlo estimate of the probability that the model would generate the correct final answer if the reasoning process were terminated at the $i$-th thinking pattern. Our analysis reveals that a non-trivial amount of intermediate segments exhibit high $p_i$ values, indicating that the model frequently produces useful reasoning components even within incorrect responses. These findings underscore the potential of selectively terminating the reasoning process at appropriate points to recover correct answers.

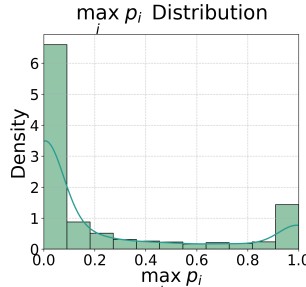

Figure 4: **Distribution of** $\max_i p_i$ **in incorrect responses of DeepScaleR-1.5B-Preview.**

### A.2    Implementation Details

For all training-based methods, the batch size was fixed at 128, and we trained the model using four NVIDIA RTX A6000 GPUs. The value of $\beta$ in Equation (13) is set to 10.0, and the ratio $\gamma/\beta$ is set to 0.3. For DeepSeek-R1-Distill-Qwen-1.5B [9] and DeepScaleR-1.5B-Preview [18], we set the sampling parameters to a temperature of 0.6, top_p of 0.95, top_k of -1, and a maximum generation length of 8192 tokens. For the Qwen2.5-Math Instruct model [37], we use the same configuration. However, due to a limitation in the SGLang framework [45], which raises an error when the maximum number of tokens exceeds 4096, we adjusted our settings accordingly.

## A.3 Algorithm of DTO

We provide a summary of the full optimization procedure in Algorithm 1.

---

**Algorithm 1 DTO**: **D**ynamic **T**hinking pattern **O**ptimization

---

**Require:** $x$: input problem, $y$: reasoning trajectory from LRM $\pi_\theta$, $a^*$: ground-truth answer, $T$: correctness threshold, $M$: samples for exit prediction, $K$: completions for finalize, $\mu_\phi$: auxiliary LLM
1: Segment $y$ into thinking patterns $[\delta_1, \ldots, \delta_{n_y}]$
2: $i' \leftarrow n_y$
3: **for** $i = 1$ to $n_y$ **do**
4:     $\tau_i \leftarrow \delta_1 \oplus \cdots \oplus \delta_i \oplus \delta_{\text{exit}}$            ▷ Construct partial trajectory by appending exit thinking pattern
5:     Sample $\mathcal{R}_i = \{r_1, \ldots, r_M\}$, $r_j \sim \pi_\theta(\tau_i)$                 ▷ Monte Carlo sampling
6:     $p_i \leftarrow |\{r \in \mathcal{R}_i \mid a^* \in r\}|/|\mathcal{R}_i|$                  ▷ Estimate correctness probability
7:     **if** $p_i \geq T$ **then**
8:         $i' \leftarrow i$
9:         **break**
10:     **end if**
11: **end for**
12: $f(\delta_i) = 1$ if $i \leq i'$, else 0                         ▷ Define binary selection function $f(\cdot)$
13: $\Delta_x^f \leftarrow [\delta_i \in y \mid f(\delta_i) = 1]$
14: Define $\delta_{\text{finalize}} \leftarrow$ "Hmm, I think this is enough to derive the final answer."
15: $\mathcal{S}_{\Delta_x^f} \leftarrow \{s_1, \ldots, s_K\}$, $s_j \sim \pi_\theta(\Delta_x^f \oplus \delta_{\text{finalize}})$             ▷ Sample completions
16: Select shortest $s^* \in \mathcal{S}_{\Delta_x^f}$ such that $a^* \in s^*$
17: $\tilde{\Delta}_x^f \leftarrow \Delta_x^f \oplus \delta_{\text{finalize}} \oplus s^*$
18: **for** each $\delta_i \in \tilde{\Delta}_x^f$ **do**
19:     Query the utility of $\delta_i$ based on the $\mu_\phi$'s evaluation conditioned on the context of $x$ and $a^*$
        (see Appendix A.4)
20:     Construct $\tilde{\Delta}_x^{f \backslash \delta_i} = \left[ \delta_j \in \tilde{\Delta}_x^f \mid \delta_j \neq \delta_i, \text{ and before "\boxed"} \right]$
21:     $g(\delta_i) \leftarrow 1$ if $\delta_i$ is deemed redundant by $\mu_\phi$ **and** decoding from $\tilde{\Delta}_x^{f \backslash \delta_i}$ yields $a^*$
22: **end for**
23: $\Delta_x^g \leftarrow [\delta_j \in \tilde{\Delta}_x^f \mid g(\delta_j) \neq 1]$
24: **return** dynamically optimized reasoning trajectory $\Delta_x^g$

---

## A.4 Prompts

---

**Default Prompt**

Please reason step by step, and put your final answer within \boxed{}.
{{ problem }}

---

**Fast Prompt**

Please reason step by step, and put your final answer within \boxed{}.
Solve the problem as quickly as possible.
{{ problem }}

---

**Prompt for the Auxiliary LLM ($\mu_\phi$ in Section 3)**

You will be given:

1. A math problem
2. A ground-truth answer
3. A series of reasoning chunks

Your task has two parts:
**STEP 1: Filtering** Decide for each chunk whether to KEEP AS IS or REMOVE. KEEP AS IS if the chunk contributes meaningfully:

- Narrowing down possibilities
- Showing intermediate calculations
- Offering partial insight (even if incomplete)
- Reinforcing understanding through repetition

REMOVE if:

- It adds no meaningful content
- It is factually incorrect in a harmful way
- It is entirely off-topic or unhelpfully redundant

**STEP 2: Final Reasoning Construction** Using only the KEPT chunks:

- Concatenate them with minimal edits for flow.
- Use each kept chunk verbatim.
- Do not paraphrase or skip ideas.
- Preserve the original tone (informal, exploratory, etc).

End the reasoning with the final answer in this format: \boxed{{...}}
**Output Format:**

```
[OUTPUT]

[CHUNK_FILTERING_RESULTS]
[Chunk 1] KEEP AS IS
[Chunk 2] REMOVE
...
[Chunk N] KEEP AS IS

[FINAL_RECONSTRUCTED_REASONING]
"""
<Clean explanation using only the kept chunks. Minimal edits for flow.
End with the boxed final answer.>
"""
```

Now here is your input:

```
### Problem:
{problem}

### Ground-Truth
{ground_truth_answer}

### Reasoning Chunks:
{thinking_patterns}
```

---

# B    Qualitative Analysis

We selected a question from the AMC2023 dataset to qualitatively compare the reasoning trajectories of the baseline model and ours. The question is "What is the degree measure of the acute angle formed by lines with slopes 2 and $\frac{1}{3}$?", and the correct answer is 45.

## B.1    A Reasoning Trajectory of the Baseline (1790 tokens)

```
<think>
Okay, so I need to find the acute angle between two lines with slopes 2 and 1/3. Hmm, I remember there's a
formula for the angle between two lines based on their slopes. Let me try to recall.

I think the formula involves the arctangent of the difference in slopes divided by 1 plus the product of the
slopes. So, if the slopes are m1 and m2, the angle \u03b8 between them is given by:

\u03b8 = arctan(|(m2 - m1)/(1 + m1*m2)|)

Is that right? Let me check. Yeah, I think that's correct because the tangent of the angle between two lines
is equal to the absolute value of (m2 - m1)/(1 + m1*m2). So, this formula gives the angle between them.

Alright, so let's assign m1 = 2 and m2 = 1/3. Plugging these into the formula:

\u03b8 = arctan(|(1/3 - 2)/(1 + (2)*(1/3))|)

First, let's compute the numerator: 1/3 - 2. That's equal to 1/3 - 6/3 = (-5)/3.

Then, the denominator: 1 + (2)*(1/3) = 1 + 2/3 = 5/3.

So, putting it together, we have:

\u03b8 = arctan(|(-5/3)/(5/3)|) = arctan(|-1|) = arctan(1)

Because the absolute value of -1 is 1. So, arctan(1) is... 45 degrees, right? Since tan(45\u00b0) = 1.

Wait, but hold on, the question asks for the acute angle. So, 45 degrees is already acute, since it's less
than 90 degrees. So, is that the answer? It seems straightforward.

But let me make sure I didn't make any mistakes in my calculations. Let's go through it again.

Slope m1 = 2, m2 = 1/3.

Difference in slopes: m2 - m1 = 1/3 - 2 = -5/3.

Product of slopes: m1*m2 = 2*(1/3) = 2/3.

Denominator: 1 + 2/3 = 5/3.

So, the argument inside arctan is |(-5/3)/(5/3)| = | -1 | = 1.

So, arctan(1) is 45 degrees. That seems correct.

Wait a second, but I remember that the angle between two lines can also be found using the formula:

tan\u03b8 = |(m2 - m1)/(1 + m1*m2)|

Which is exactly what I used. So, that's consistent.

But just to double-check, let me visualize. If one line has a slope of 2, that's pretty steep, going up 2
units for every 1 unit to the right. The other line has a slope of 1/3, which is a gentle slope, going up 1
unit for every 3 units to the right.

So, if you imagine these two lines intersecting, the angle between them is likely to be acute because both
slopes are positive but one is steeper. So, 45 degrees is reasonable.

Wait, 45 degrees is a common angle, so that seems plausible.

Alternatively, I can think of the angle each line makes with the x-axis and then find the difference between
those two angles.

The angle \u03b1 that a line with slope m makes with the x-axis is given by \u03b1 = arctan(m). So, for m1 =
2, \u03b11 = arctan(2), and for m2 = 1/3, \u03b12 = arctan(1/3).

Then, the angle between the two lines would be |\u03b11 - \u03b12|. Since both \u03b11 and \u03b12 are
positive angles less than 90 degrees (since slopes are positive), the difference will also be positive, and
since we're taking the absolute value, it'll give the acute angle.

Let me compute that way to verify.

First, compute \u03b11 = arctan(2). Let me approximate that. I know that arctan(1) is 45\u00b0,
arctan(\u221a3) is 60\u00b0, so arctan(2) is somewhere between 60\u00b0 and 75\u00b0, maybe around
63.4349\u00b0.

Similarly, arctan(1/3). Since 1/3 is less than 1, arctan(1/3) is approximately 18.4349\u00b0.

So, the difference is 63.4349\u00b0 - 18.4349\u00b0 = 45\u00b0. So, that confirms it.

Therefore, the acute angle between the two lines is 45 degrees.
```

```
Wait, so both methods give the same answer. That gives me more confidence.

Alternatively, I can think about the tangent of the angle between them, which we found was 1, so arctan(1)
is 45\u00b0, so that's consistent.

So, whether I use the formula with the slopes or compute the angles each line makes with the x-axis and
subtract them, I get 45 degrees. Therefore, I think that's the correct answer.

I don't see any mistakes in my reasoning. The key steps are:

1. Recognize the formula for the angle between two lines: \u03b8 = arctan(|(m2 - m1)/(1 + m1*m2)|)
2. Plug in the slopes m1 = 2 and m2 = 1/3
3. Calculate the numerator and denominator
4. Simplify the expression inside arctan
5. Find that it equals arctan(1) = 45\u00b0
6. Confirm with another method, like computing the individual angles and subtracting them

All steps check out, so I'm pretty sure 45 degrees is the correct answer.

**Final Answer**
The acute angle formed by the lines is \\boxed{45} degrees.</think>

To find the acute angle formed by lines with slopes \\(2\\) and \\(\\frac{1}{3}\\), we use the formula for
the angle \\(\\theta\\) between two lines with slopes \\(m_1\\) and \\(m_2\\):

\\[
\\theta = \\arctan\\left(\\left|\\frac{m_2 - m_1}{1 + m_1 m_2}\\right|\\right)
\\]

Assigning \\(m_1 = 2\\) and \\(m_2 = \\frac{1}{3}\\):

1. Compute the numerator:
\\[
m_2 - m_1 = \\frac{1}{3} - 2 = \\frac{1}{3} - \\frac{6}{3} = -\\frac{5}{3}
\\]

2. Compute the denominator:
\\[
1 + m_1 m_2 = 1 + 2 \\cdot \\frac{1}{3} = 1 + \\frac{2}{3} = \\frac{5}{3}
\\]

3. Simplify the argument inside the arctangent:
\\[
\\left| \\frac{-\\frac{5}{3}}{\\frac{5}{3}} \\right| = \\left| -1 \\right| = 1
\\]

4. Therefore, the angle \\(\\theta\\) is:
\\[
\\theta = \\arctan(1) = 45^\\circ
\\]

To verify, we can also compute the angles each line makes with the x-axis and find the difference. The
angles are:
- \\(\\alpha_1 = \\arctan(2) \\approx 63.4349^\\circ\\)
- \\(\\alpha_2 = \\arctan\\left(\\frac{1}{3}\\right) \\approx 18.4349^\\circ\\)

The difference is:
\\[
63.4349^\\circ - 18.4349^\\circ = 45^\\circ
\\]

Thus, the acute angle formed by the lines is \\(\\boxed{45}\\) degrees.
```

## B.2  A Reasoning Trajectory of the Ours (814 tokens)

```
<think>
Okay, so I need to find the acute angle between two lines with slopes 2 and 1/3. Hmm, I remember that
there's a formula involving the slopes of two lines to find the angle between them. Let me think. I think
it's something with the arctangent of the difference of the slopes divided by something else... Maybe it's
tan(theta) equals the absolute value of (m2 - m1)/(1 + m1*m2). Yeah, that sounds familiar from my geometry
class.

Alright, so let me write that down. The formula is tan(theta) = |(m2 - m1)/(1 + m1*m2)|. So here, m1 is 2
and m2 is 1/3. Plugging those in, it should be tan(theta) equals the absolute value of (1/3 - 2)/(1 +
(2)(1/3)). Let me compute numerator and denominator separately.

First, the numerator: 1/3 - 2. Hmm, I need to subtract these. Let me convert 2 into thirds to make it
easier. 2 is equal to 6/3. So 1/3 - 6/3 is -5/3. Okay, so numerator is -5/3.

Denominator: 1 + (2)(1/3). Let me compute that. 2 times 1/3 is 2/3. Then, adding 1, which is 3/3, so 2/3 +
3/3 is 5/3. So denominator is 5/3.

Therefore, tan(theta) equals the absolute value of (-5/3)/(5/3). When you divide those fractions, the 5/3
cancels out, leaving | -1 |. So tan(theta) is 1.

Now, if tan(theta) is 1, then theta is arctangent of 1. I remember that tan(45 degrees) is 1. So theta is 45
degrees. Hmm, since it's asking for the acute angle, which is less than 90 degrees, and 45 degrees is acute,
so that should be the answer.
```

```
I think I did everything right. Let me recap: slopes are 2 and 1/3. Applied the formula, computed numerator
as negative, denominator positive, got tan(theta)=1, theta is 45 degrees. Yeah, sounds good.

**Final Answer**
The acute angle formed by the lines is \\boxed{45} degrees.
</think>

To find the acute angle formed by lines with slopes \\(2\\) and \\(\\frac{1}{3}\\), we use the formula for the
the tangent of the angle between two lines:

\\[
\\tan(\\theta) = \\left| \\frac{m_2 - m_1}{1 + m_1 m_2} \\right|
\\]

Here, \\(m_1 = 2\\) and \\(m_2 = \\frac{1}{3}\\). Plugging these values into the formula, we get:

\\[
\\tan(\\theta) = \\left| \\frac{\\frac{1}{3} - 2}{1 + 2 \\cdot \\frac{1}{3}} \\right|
\\]

First, compute the numerator:

\\[
\\frac{1}{3} - 2 = \\frac{1}{3} - \\frac{6}{3} = -\\frac{5}{3}
\\]

Next, compute the denominator:

\\[
1 + 2 \\cdot \\frac{1}{3} = 1 + \\frac{2}{3} = \\frac{3}{3} + \\frac{2}{3} = \\frac{5}{3}
\\]

Thus, we have:

\\[
\\tan(\\theta) = \\left| \\frac{-\\frac{5}{3}}{\\frac{5}{3}} \\right| = \\left| -1 \\right| = 1
\\]

Since \\(\\tan(\\theta) = 1\\), we find that \\(\\theta = 45^\\circ\\).

\\[
\\boxed{45}
\\]
```

## C   Datasets

The information about the dataset we used is as follows:

- **MATH [10]:** https://github.com/hendrycks/math. The dataset is released under the MIT license, and we adopt the same data split as used in Luo et al. [17].
- **GSM8K [5]:** https://huggingface.co/datasets/openai/gsm8k. The dataset is released under the MIT license.
- **Gaokao [43]:** https://github.com/OpenLMLab/GAOKAO-Bench. The dataset is released under the Apache-2.0 license, and we adopt the same data split as used in Luo et al. [17].
- **AMC:** https://huggingface.co/datasets/zwhe99/amc23.
- **AIME:** https://huggingface.co/datasets/AI-MO/NuminaMath-CoT.
- **MMLU-Pro [33]:** https://huggingface.co/datasets/TIGER-Lab/MMLU-Pro. The dataset is released under the Apache-2.0 license.

## D   Limitations

While our method demonstrates consistent improvements in reasoning efficiency and accuracy, it has several limitations. First, the effectiveness of our approach has been primarily validated on mathematical and reasoning-focused benchmarks. While we validated our method on other domains using the MMLU-Pro dataset in Section 4.3, extending the evaluation to broader domains, such as open-ended tasks, remains a promising direction for future work. Second, the approach assumes access to multiple model-generated responses per problem, which may not always be feasible in highly resource-constrained settings. Investigating ways to reduce the computational overhead of the optimization phase itself could further improve the practicality of the approach in real-world deployments.

## E  Broader Impacts

This work aims to enhance the reasoning efficiency of Large Reasoning Models (LRMs) by optimizing their internal decision-making processes. By reducing unnecessary reasoning steps while maintaining or improving accuracy, the proposed method has the potential to substantially lower inference cost and latency. This can benefit downstream applications by enabling more computationally efficient and environmentally sustainable deployment of LRMs, particularly in resource-constrained settings. However, if LRMs are misused–for instance, to generate plausible fake information, automate biased decision-making, or produce large-scale deceptive content–they may contribute to negative societal impacts. It is therefore important to remain mindful of the potential for such technologies to be applied in harmful ways, even when developed with efficiency and performance in mind.

