# OpenReview forum: "Don’t Think Longer, Think Wisely: Optimizing Thinking Dynamics for Large Reasoning Models"
_NeurIPS.cc/2025/Conference — NeurIPS 2025 poster_

### Official Review · Reviewer_z6Kh · 2025-06-03

**Clarity:** 3
**Significance:** 3
**Originality:** 3
**Rating:** 4
**Confidence:** 2

**Summary:**

This paper tackles the "overthinking" issue in Large Reasoning Models (LRMs), where they produce overly long and complex reasoning, harming efficiency. The authors propose a Dynamic Thinking pattern Optimization (DTO) framework. This framework aims to segment reasoning paths into "thinking patterns," then identify and promote useful ones while removing those that are detrimental. The DTO process involves finding good stopping points for reasoning using Monte Carlo estimation and then pruning unhelpful segments with an auxiliary Large Language Model (LLM). The authors further refine LRMs by using preference optimization, creating a dataset of DTO-improved trajectories versus unoptimized ones to guide the model. Experiments across mathematical reasoning tasks show their method can reduce computational load (like FLOPs and token count) while keeping or even boosting accuracy. For instance, it reduced attention FLOPs by up to 47% while maintaining accuracy for initially correct responses, and improved accuracy by 15.6% for some initially incorrect ones. With preference optimization, they report up to a 12% accuracy gain and a token reduction from about 5,000 to 3,000.

**Questions:**

See weakness

**Ethical Concerns:**

["NO or VERY MINOR ethics concerns only"]

**Final Justification:**

In my original review, I raised concerns about the significance of DTO’s improvements and whether some experimental settings—particularly the fixed 8192-token truncation—might unfairly disadvantage models that naturally produce longer reasoning chains. I also questioned whether simpler baselines, like forced-finalization, were considered, and whether the method generalized beyond small-scale models.

The rebuttal addressed these points with thoughtful experiments, including ablations, extended token limits, and stronger baselines. I found these responses helpful and well-executed.

However, my overall assessment remains cautious due to the lack of rigor in the original submission. Some of the evaluation setups could have been more balanced and better justified in the main paper. As a result, I have increased my score to reflect the strength of the rebuttal, but slightly lowered my confidence due to concerns about the paper's initial rigor.

**Limitations:**

yes

**Quality:**

3

**Strengths And Weaknesses:**

I think the paper is reasonably spund. The approach of optimizing thinking dynamics in LRMs by segmenting reasoning into "thinking patterns" and then refining them is quite interesting. The empirical results, showing reductions in computational overhead like FLOPs and token usage while often maintaining or even improving accuracy, appear promising. I particularly noted the ability of the DTO framework to sometimes correct initially flawed reasoning paths, which is a valuable outcome. The reported efficiency gains, such as reducing attention FLOPs by up to 47% and token usage from around 5,000 to 3,000 while also achieving accuracy improvements, are significant.

However, I have a few points:

1.  It seems the DTO framework has two main stages: identifying an optimal termination point using Monte Carlo estimation, and then pruning redundant segments with an auxiliary LLM. I'm wondering if there was an ablation study performed to understand the individual contribution of each of these stages. It would be insightful to know which component contributes more to the overall improvement, or if both are equally critical.

2.  The maximum response length was set to 8192 tokens for the experiments. I'm a bit concerned whether many of the baseline models might not have fully generated their responses within this limit, potentially truncating their reasoning, leading to incorrect answer (which is a super critical problem). This could affect the comparison, especially if the DTO-optimized models inherently produce shorter responses.

3.  Based on my experience, smaller models like 1.5B or even 7B parameters can sometimes (even often) get stuck in loops or fail to finalize their answers. I think a simple but potentially strong baseline could involve forcing the model to generate an answer once it reaches a certain token limit, perhaps using a similar "finalization" mechanism as described in the paper. This might offer a more direct comparison for the conciseness and termination aspects of DTO.

4. While I understand compute limitations might restrict extensive experiments on much larger models, I do have some slight reservations about the generalizability of these findings to significantly larger models without further evidence (considering small models' may have limited "self-correction" ability).

---

> ### Author Rebuttal · Authors · 2025-07-31
>
> **[W1]**
>
> > **Reviewer’s Comment**: An ablation study to understand the individual contribution of each of these stages would be insightful.
>
> Thank you for raising this valuable point. To better understand the contribution of each stage, we conducted an ablation study that incrementally introduces each component of our framework:
>
> 1. **DTO (Stage 1)**: We first add the mechanism that identifies the appropriate point at which reasoning should be finalized.
> 2. **DTO (Stage 1 + 2)**: Building on top of Stage 1, we incorporate a module that prune intermediate thinking patterns that do not meaningfully contribute to the reasoning objective.
>
> We used DeepSeek-R1-Distill-Qwen-1.5B as the base model for this study, and the results are presented in the table below. As shown, both stages independently yield meaningful improvements. Notably, Stage 1 alone provides substantial gains in reasoning efficiency by enabling early termination once sufficient evidence has been gathered. While the additional gains from Stage 2 are more moderate, we can observe that it still brings meaningful improvements. The full DTO (Stage 1 + 2) consistently delivers the strongest results, outperforming all baselines in terms of efficiency while maintaining or even improving accuracy, as reported in Tables 1 and 2 of the main paper.
>
> We will include the full ablation results and analysis in the revised version.
>
> | Dataset |  | Base | DTO (Stage 1) | DTO (Stage 1 + 2) |
> | --- | --- | --- | --- | --- |
> | **MATH** | ACC | 79.80 | 85.21 | 85.48 |
> |  | #Tokens | 3543.44 | 2256.26 | 1936.19 |
> |  | Eff. | 1.000 | 1.677 | **1.960** |
> | **GSM8K** | ACC | 82.13 | 83.95 | 83.91 |
> |  | #Tokens | 1382.99 | 1050.54 | 844.18 |
> |  | Eff. | 1.000 | 1.346 | **1.674** |
> | **Gaokao** | ACC | 66.62 | 72.79 | 72.66 |
> |  | #Tokens | 3725.16 | 2559.51 | 2137.59 |
> |  | Eff. | 1.000 | 1.590 | **1.901** |
> | **AMC2023** | ACC | 58.25 | 70.00 | 70.25 |
> |  | #Tokens | 5338.54 | 3768.81 | 3376.98 |
> |  | Eff. | 1.000 | 1.702 | **1.907** |
> | **AIME2024** | ACC | 21.44 | 30.00 | 28.00 |
> |  | #Tokens | 7359.24 | 6301.40 | 5877.44 |
> |  | Eff. | 1.000 | 1.634 | **1.635** |
> | **AIME2025** | ACC | 18.89 | 21.67 | 21.11 |
> |  | #Tokens | 7236.66 | 6237.63 | 5689.38 |
> |  | Eff. | 1.000 | 1.331 | **1.421** |
>
> **[W2]**
>
> > **Reviewer’s Comment**: Because the evaluation was conducted under an 8192-token limit, there is a possibility that some responses were truncated due to this length constraint.
>
> Thank you for raising this concern regarding the potential truncation of responses. We would like to firstly clarify that **all models were evaluated under the same maximum response length of 8192 tokens**, ensuring a fair and consistent comparison across methods.
>
> To further examine whether this constraint may have disadvantaged baseline methods, we conducted additional experiments under a **higher token limit of 16,384**, using DeepSeek-R1-Distill-Qwen-1.5B as the base model. As shown in the tables below, we observe that all methods exhibit a general increase in both accuracy and token usage when granted more space to complete their reasoning. This is expected, as longer context allows models to better develop and finalize their solutions.
>
> However, **DTO continues to outperform** both the base model and the strongest baseline (FCS + Ref.), even under this more generous constraint. In particular, DTO achieves the **highest accuracy while using the fewest tokens**, leading to the best efficiency scores overall. For example, on AMC2023, DTO improves accuracy by **8.2 points over the baseline**, while reducing token usage by over 50%, resulting in an efficiency gain of **2.43x**. Similar patterns are observed on Gaokao and AIME2024.
>
> These findings confirm that DTO’s effectiveness is **not merely a byproduct of producing shorter responses** under certain token limits, but rather reflects **genuine improvements in reasoning quality efficiency**, even when models are allowed more room to reason.
>
> | Dataset |  | Baseline | FCS + Ref. | DTO (Ours) |
> | --- | --- | --- | --- | --- |
> | **Gaokao** | ACC | 69.35 | 72.92 | 73.12 |
> |  | #Tokens | 4845.63 | 3214.31 | 2277.71 |
> |  | Eff. | 1.000 | 1.585 | **2.243** |
> | **AMC2023** | ACC | 66.88 | 73.75 | 75.00 |
> |  | #Tokens | 8051.86 | 5648.81 | 3712.44 |
> |  | Eff. | 1.000 | 1.572 | **2.432** |
> | **AIME2024** | ACC | 26.67 | 27.50 | 30.00 |
> |  | #Tokens | 12345.34 | 9756.87 | 7871.91 |
> |  | Eff. | 1.000 | 1.305 | **1.764** |
>
> **[W3]**
>
> > **Reviewer’s Comment**: A simple yet potentially strong baseline could involve forcing the model to generate an answer once it reaches a certain token limit.
>
> Thank you for the thoughtful suggestion. We agree that smaller models such as 1.5B or 7B parameters may occasionally struggle to finalize their reasoning and that encouraging termination at a specific token threshold can serve as a useful baseline for evaluating our method.
>
> To assess this idea, we implemented the proposed baseline: for a range of maximum token limits (1024, 2048, 4096, and 6144), we prompted the base model, DeepSeek-R1-Distill-Qwen-1.5B, to produce a final boxed answer when nearing the token limit. To facilitate finalization, we appended the following instruction: *"Wait, I suddenly got the final answer to the whole problem.\n\nFinal Answer:"*.
>
> This prompt-based mechanism was applied to the base model under each token constraint, simulating a forced finalization strategy similar in spirit to the one described in our paper.
>
> As shown in the results, we observe a consistent trend: **as the token limit increases, both accuracy and the number of generated tokens increase**, indicating that longer sequences enable more complete reasoning. However, we can observe that **our method, DTO, achieves higher accuracy while using fewer or comparable tokens**, highlighting its strength in both **reasoning quality and efficiency**.
>
> We will include the full result table in the revised version.
>
> | Dataset |  | Baseline + 1024 | Baseline + 2048 | Baseline + 4096 | Baseline + 6144 | DTO (Ours) |
> | --- | --- | --- | --- | --- | --- | --- |
> | **MATH** | ACC | 63.59 | 74.78 | 78.55 | 79.97 | 85.48 |
> |  | #Tokens | 1009.65 | 1783.91 | 2642.43 | 3124.27 | 1936.19 |
> | **GSM8K** | ACC | 79.40 | 80.48 | 79.78 | 80.12 | 83.91 |
> |  | #Tokens | 576.62 | 722.42 | 850.67 | 928.66 | 844.18 |
> | **Gaokao** | ACC | 50.06 | 57.86 | 65.26 | 67.86 | 72.66 |
> |  | #Tokens | 957.04 | 1661.44 | 2551.03 | 3092.08 | 2137.59 |
> | **AMC2023** | ACC | 32.50 | 50.00 | 67.50 | 68.12 | 70.25 |
> |  | #Tokens | 1022.44 | 1965.05 | 3416.82 | 4385.90 | 3376.98 |
> | **AIME2024** | ACC | 3.33 | 14.17 | 20.83 | 24.17 | 28.00 |
> |  | #Tokens | 1024.00 | 2048.00 | 4009.87 | 5686.45 | 5877.44 |
> | **AIME2025** | ACC | 1.67 | 12.50 | 17.50 | 18.33 | 21.11 |
> |  | #Tokens | 1024.00 | 2048.00 | 3986.50 | 5715.72 | 5689.38 |
>
> **[W4]**
>
> > **Reviewer’s Comment**: There is some concern regarding the generalizability of the findings to larger models.
>
> Thank you for raising this important question regarding the generalizability of our method to larger language reasoning models. To address this point, we conducted additional experiments within our available resources to empirically evaluate DTO’s performance in larger-scale settings.
>
> Specifically, we tested our method on a **larger model, i.e., DeepSeek-R1-Distill-Qwen-7B,** and considered three different settings on this model:
>
> - the base model,
> - the strongest baseline (**FCS + Ref.**), and
> - our proposed method (**DTO**).
>
> These experiments followed the same training and evaluation protocols as outlined in the main paper, except for one change: due to computational constraints, we used **2,500 training instances** instead of 5,000. All other experimental settings remained consistent to ensure comparability. Additionally, please note that, due to the same computational constraints, we did not perform hyperparameter tuning and instead used the default configuration from the main paper.
>
> As shown in the extended results table below, **DTO continues to achieve superior efficiency metrics** even at the 7B scale. It **outperforms both the base model and the FCS + Ref. baseline, which is the strongest baseline,** across all benchmarks, delivering improvements in both accuracy and token efficiency. These findings suggest that the **benefits of DTO generalize beyond 1.5B-sized models** and that the method is robust to increased model capacity.
>
> We sincerely appreciate your suggestion, and we believe that these additional results strengthen the case for the applicability of DTO to larger reasoning models. We will incorporate these findings into the revised version of our main paper.
>
> | Dataset |  | Baseline | FCS + Ref. | DTO (Ours) |
> | --- | --- | --- | --- | --- |
> | **MATH** | ACC | 90.01 | 90.83 | 92.63 |
> |  | #Tokens | 3288.73 | 2756.03 | 2339.03 |
> |  | Eff. | 1.000 | 1.204 | **1.447** |
> | **GSM8K** | ACC | 92.32 | 92.59 | 93.03 |
> |  | #Tokens | 1731.13 | 1356.28 | 1148.84 |
> |  | Eff. | 1.000 | 1.280 | **1.518** |
> | **Gaokao** | ACC | 78.57 | 79.87 | 81.30 |
> |  | #Tokens | 3402.44 | 2924.27 | 2519.02 |
> |  | Eff. | 1.000 | 1.183 | **1.398** |
> | **AMC2023** | ACC | 78.75 | 82.50 | 86.25 |
> |  | #Tokens | 4885.61 | 4356.48 | 3731.45 |
> |  | Eff. | 1.000 | 1.175 | **1.434** |
> | **AIME2024** | ACC | 37.50 | 43.33 | 45.00 |
> |  | #Tokens | 6803.64 | 6724.27 | 6047.72 |
> |  | Eff. | 1.000 | 1.169 | **1.350** |
> | **AIME2025** | ACC | 30.00 | 32.50 | 33.33 |
> |  | #Tokens | 7013.32 | 6580.07 | 6548.10 |
> |  | Eff. | 1.000 | 1.155 | **1.190** |

---

> > ### Comment · Reviewer_z6Kh · 2025-08-05
> >
> > Thank you to the authors for the detailed responses and the additional experiments. I appreciate the new ablation studies, the analysis under extended token limits, and the inclusion of forced-finalization baselines, which address my initial concerns and significantly improve the clarity of the paper.
> >
> > That said, I still have a few reservations. First, using a fixed 8192-token cutoff across all models may not be entirely fair, as it can disproportionately penalize models that naturally produce longer reasoning chains. I suggest either relaxing this constraint in some evaluations or more explicitly discussing its implications in the original paper. Second, in the 6144-token forced-finalization experiments on AIME24 and AIME25, DTO does not appear to reduce token usage compared to the baseline (e.g., 5686.45 vs 5877.44 on AIME24). This makes it harder to prove a clear efficiency gain. I encourage the authors to highlight DTO's token-saving benefits at higher performance levels, where the method’s advantages may be more compelling.
> >
> > Above all, I will raise my score if certain valuable experiments can be incorporated into the main paper.

---

### Official Review · Reviewer_LB6E · 2025-06-29

**Clarity:** 3
**Significance:** 2
**Originality:** 3
**Rating:** 4
**Confidence:** 4

**Summary:**

This paper proposes a response optimization method for LRMs (large reasoning models). Specifically, it first uses Monte Carlo sampling to truncate responses and then employs an auxiliary LLM to remove redundant segments, aiming to reduce length without compromising accuracy. Preference optimization based on the optimized responses achieves competitive efficiency.

**Questions:**

1. I'm confused about how DTO is applied to incorrect responses, as discussed in Sections 3.3 and Section 4.1.
    1. In Section 3.3, DTO relies on the gold answer in three ways: to truncate the given response, to get the shortest correct completion, and to remove redundant segments. Then the optimized response should achieve 100% accuracy if a correct trajectory exists in the sampling space. Therefore, Lines 192-196 are *not interesting*, as this is an unfair comparison with the initially yielded responses.
    2. In Section 4.1 Lines 216-217, it is not guarenteed that applying DTO to a randomly sampled incorrect response will lead to an optimized response. How do you handle this?
2. How do you segment a response y into a sequence of distinct thinking patterns? How many segments does a response have in average? And calculating all of these, how many completions need to be sampled per response to get an optimized response using DTO?
3. FCS employs QwQ-32B. Can your proposed method show efficiency in larger LRMs, at least on 7B models?

**Ethical Concerns:**

["NO or VERY MINOR ethics concerns only"]

**Final Justification:**

I think using DTO to construct preference data is a reasonable and promising idea. While the improvements (e.g., saving a few hundred tokens in long responses) are relatively modest, the experiments are solid and the method could inspire future work in this direction.

**Limitations:**

yes

**Paper Formatting Concerns:**

No.

**Quality:**

3

**Strengths And Weaknesses:**

**Strengths**

1. The idea of optimizing the individual contributions of reasoning segments is insightful, and the authors make a clear effort to achieve this goal through their proposed method.
2. The experimental results are competitive and the statistical analysis is well-executed and easy to appreciate.
3. The paper is content-rich, and the method section is clearly described.

**Weaknesses**

1. The proposed method is less efficient in collecting training samples compared to several baselines like FCS.
2. The experiments are limited to two 1.5B models, making it unclear whether the proposed method remains effective on larger LRMs.

---

> ### Author Rebuttal · Authors · 2025-07-31
>
> **[W1 & Q2]**
> > **Reviewer’s Comment**: The method seems less efficient in collecting training samples. How is the response segmented into thinking patterns, and how many completions are needed per response to optimize it using DTO?
>
> As described in Section 3 of the main paper, we segment the reasoning trajectory $y$ into distinct thinking patterns by leveraging discourse cues such as *“Wait”* or *“Alternatively”*, following prior work [1, 2]. These expressions commonly emerge in large reasoning models' outputs and are known to mark shifts in the model’s reasoning process, including transitions into self-verification, intermediate summarization, or revised hypotheses.
>
> Empirical analysis reveals that a typical reasoning trajectory contains approximately 22 thinking patterns on average. However, not all of these require decoding. For the first stage of DTO - which identifies the point at which the reasoning should be finalized - we evaluate the probability of producing the correct answer from each partial trajectory $\tau_i = \delta_1 \oplus \cdots \oplus \delta_i \oplus \delta_{\text{exit}}$, as defined in Equation (6). We then determine the earliest index $i'$ at which the probability surpasses a predefined confidence threshold. On average, this threshold is reached around the 14th segment, which implies that only the remaining 8 chunks typically require decoding.
>
> Altogether, following the setup in Section 3.3, DTO requires approximately 92 completions per response. Importantly, these completions are lightweight, since the decoding is limited to a small number of tokens - just 16 more than the length of the ground-truth answer- and can be efficiently parallelized using fast inference frameworks [3, 4]. Additionally, while DTO may involve a higher sample collection cost than some baselines, such as FCS + Ref., it is important to note that this procedure is performed offline during training. Once trained, the model yields substantial improvements in inference-time reasoning efficiency, as demonstrated in Tables 1 and 2 of the main paper. Therefore, we believe that the one-time cost of training is justifiable given the long-term benefits in runtime efficiency and scalability.
>
> We will update the manuscript to clarify these aspects more explicitly. Thank you again for your valuable feedback.
>
> **[W2 & Q3]**
> > **Reviewer’s Comment**: Can your proposed method show efficiency in larger LRMs, at least on 7B models?
>
> Thank you for your thoughtful question regarding the scalability of our method to larger language reasoning models. To address this point, we conducted additional experiments within our available resources to empirically evaluate DTO’s performance in larger-scale settings.
>
> Specifically, we tested our method on a **larger model, i.e., DeepSeek-R1-Distill-Qwen-7B,** and considered three different settings on this model:
>
> - the base model,
> - the strongest baseline (**FCS + Ref.**), and
> - our proposed method (**DTO**).
>
> These experiments followed the same training and evaluation protocols as outlined in the main paper, except for one change: due to computational constraints, we used **2,500 training instances** instead of 5,000. All other experimental settings remained consistent to ensure comparability. Additionally, please note that, due to the same computational constraints, we did not perform hyperparameter tuning and instead used the default configuration from the main paper.
>
> As shown in the extended results table below, **DTO continues to achieve superior efficiency metrics** even at the 7B scale. It **outperforms both the base model and the FCS + Ref. baseline, which is the strongest baseline,** across all benchmarks, delivering improvements in both accuracy and token efficiency. These findings suggest that the **benefits of DTO generalize beyond 1.5B-sized models** and that the method is robust to increased model capacity.
>
> We sincerely appreciate your suggestion, and we believe that these additional results strengthen the case for the applicability of DTO to larger reasoning models. We will incorporate these findings into the revised version of our main paper.
>
> | Dataset |  | Baseline | FCS + Ref. | DTO (Ours) |
> | --- | --- | --- | --- | --- |
> | **MATH** | ACC | 90.01 | 90.83 | 92.63 |
> |  | #Tokens | 3288.73 | 2756.03 | 2339.03 |
> |  | Eff. | 1.000 | 1.204 | **1.447** |
> | **GSM8K** | ACC | 92.32 | 92.59 | 93.03 |
> |  | #Tokens | 1731.13 | 1356.28 | 1148.84 |
> |  | Eff. | 1.000 | 1.280 | **1.518** |
> | **Gaokao** | ACC | 78.57 | 79.87 | 81.30 |
> |  | #Tokens | 3402.44 | 2924.27 | 2519.02 |
> |  | Eff. | 1.000 | 1.183 | **1.398** |
> | **AMC2023** | ACC | 78.75 | 82.50 | 86.25 |
> |  | #Tokens | 4885.61 | 4356.48 | 3731.45 |
> |  | Eff. | 1.000 | 1.175 | **1.434** |
> | **AIME2024** | ACC | 37.50 | 43.33 | 45.00 |
> |  | #Tokens | 6803.64 | 6724.27 | 6047.72 |
> |  | Eff. | 1.000 | 1.169 | **1.350** |
> | **AIME2025** | ACC | 30.00 | 32.50 | 33.33 |
> |  | #Tokens | 7013.32 | 6580.07 | 6548.10 |
> |  | Eff. | 1.000 | 1.155 | **1.190** |
>
> **[Q1]**
> > **Reviewer’s Comment:** It is unclear how DTO is applied when the initial response is incorrect.
>
> Thank you for raising this question. Let us clarify the intended interpretation and provide further explanation.
>
> Firstly, the analysis in lines 192-196 is not about optimizing responses that are already correct. Rather, it highlights an important capability of our framework: even when the originally sampled response is incorrect, applying DTO can sometimes recover a correct reasoning trajectory.
>
> This behavior is particularly valuable in challenging settings where most or all sampled responses from the base reasoning model are incorrect. In such cases, prior methods such as FCS + Ref. or DAST are unable to generate preference pairs and must discard the instance entirely, as they rely on having at least one correct response in the sampled pool.
>
> In contrast, DTO enables us to rescue such difficult instances by producing a correct, optimized trajectory from an initially incorrect response. While this recovery is not guaranteed in all cases, when it does succeed, it allows us to include these cases in the pairwise training dataset, potentially offering strong learning signals in hard problem regimes. We believe this capability is one of the strengths of our approach, as reflected in the strong accuracy results reported in Tables 1 and 2 of the main paper, and we will revise the corresponding section to make this point more explicit.
>
> ### References
>
> [1] Marjanović, Sara Vera, et al. "DeepSeek-R1 Thoughtology: Let's think about LLM Reasoning." *arXiv preprint arXiv:2504.07128* (2025).
>
> [2] Lu, Ximing, et al. "Retro-search: Exploring untaken paths for deeper and efficient reasoning." *arXiv preprint arXiv:2504.04383* (2025).
>
> [3] Zheng, Lianmin, et al. "Sglang: Efficient execution of structured language model programs." Advances in neural information processing systems 37 (2024): 62557-62583.
>
> [4] Fu, Yichao, et al. "Efficiently Scaling LLM Reasoning with Certaindex." arXiv preprint arXiv:2412.20993 (2024).

---

> ### Comment · Reviewer_LB6E · 2025-08-04
>
> Thank you for your response and the additional experiments. I’d like to further clarify Q1:
>
> - For the claim “even when the originally sampled response is incorrect, applying DTO can sometimes recover a correct reasoning trajectory”, I believe this fundamentally relies on the sampling space (not full response resampling, but also process-level sampling as in [1]). Your method uses the gold answer to directly filter for correct responses from this sampling space, so the observed accuracy improvement is an expected outcome rather than a clear benefit of your proposed approach. While this does not diminish your contribution in using DTO to construct preference data, I find the analysis in Section 3.3 insufficiently convincing in isolating the method’s true impact.
>
> [1] MATH-SHEPHERD: VERIFY AND REINFORCE LLMSSTEP-BY-STEP WITHOUT HUMAN ANNOTATIONS. arXiv:2312.08935.
>
> Overall, I think using DTO to construct preference data is a reasonable and promising idea. While the improvements (e.g., saving a few hundred tokens in long responses) are relatively modest, the experiments are solid and the method could inspire future work in this direction. I will maintain my current score.

---

### Official Review · Reviewer_Nq5r · 2025-07-02

**Clarity:** 2
**Significance:** 3
**Originality:** 3
**Rating:** 4
**Confidence:** 4

**Summary:**

The authors introduce Dynamic Thinking Pattern Optimization (DTO), a mechanism designed to solve the 'overthinking' of large reasoning models (LRMs). DTO first segments a generated reasoning trace into modular thinking patterns and locates the earliest point where the model can produce the correct answer, yielding a truncated path. Then, DTO elicits a minimal answer token string, which produces an optimized trajectory by removing inefficient patterns. These optimized reasoning traces achieve reasoning efficiency by reducing attention FLOPs by up to 47%. The authors also provide preference optimization with DTO, which also enhances the accuracy performance.

**Questions:**

1) Can DTO handle queries where no golden label is available? I wonder whether DTO is applicable in such settings.

2) Can we use the same LRM as an auxiliary LLM? For instance, in this paper, can we use DeepSeek-R1-Distill-Qwen-1.5B or DeepScaleR-1.5B-Preview? If so, why didn't the authors conduct the experiments? If not, it would be helpful to explain the reason.

**Ethical Concerns:**

["NO or VERY MINOR ethics concerns only"]

**Final Justification:**

I am maintaining my evaluation as "vorderline accept" because the paper's core idea is reasonable, and the authors have committed to addressing my main concern in the final version.

- **Strengths:** The paper's core idea, the DTO framework, addresses the "overthinking" problems in large reasoning models.

- **Remaining Concern:** My main concern is the use of an oversized auxiliary LLM (70B). This makes it difficult to assess the standalone contribution of the DTO methodology, separate from the power of the large model. **However**, this concern is addressed by the authors' rebuttal and also the firm commitment to include the crucial ablation study I requested in the final version. Thus, I maintain my position to "borderline accept", contingent on this revision.

**Limitations:**

Yes.

**Paper Formatting Concerns:**

The paper follows the given format.

**Quality:**

3

**Strengths And Weaknesses:**

## Strengths

1) While current studies have focused on making longer reasoning traces, the authors' motivation to shorten and optimize the length of reasoning is fascinating. It is also intriguing to formulate this as a constrained optimization problem.

2) By delivering simultaneous gains in both efficiency and accuracy, DTO outperforms baseline methods, except for #Tokens of Instruct ver. DTO's ability to generate minimal yet sufficient reasoning paths not only lowers inference cost but also supplies high-quality supervision for further model improvement.

3) I can see that DTO is not tied to arithmetic reasoning alone, but can compact and preserve the correctness of chains-of-thought in diverse subject areas from Table 4. Although the authors mentioned mathematical-focused validation as one of their limitations, I am pretty confident that DTO would reduce the reasoning token lengths or attention FLOPs in more varied benchmarks.


## Weaknesses

1) DTO's pruning stage, after determining the 'finalize' thinking pattern, requires an auxiliary LLM, which is different from the original LRM. The auxiliary LLM judges and removes redundant thinking patterns. The authors use a 70B model for an auxiliary model, whereas the original model's size is 1.5B. I wonder whether it is fair to use the auxiliary model of such a large size, when the baselines seem to only use 1.5B, which is the same size as the original LRM. However, I do not see any justification for using the large auxiliary model for DTO.

2) DTO splits the reasoning trace with words such as 'Wait' or 'Alternatively'. DTO cannot split if such words are not contained in the reasoning trace. I presume the paper needs a more sophisticated method to segment a reasoning trajectory in a sequence of thinking patterns. If the current method is sufficient, please explain the reason.

3) Tables 1--4 contain Instruct ver. as a baseline, which is an instruction-tuned version of LRMs. I noticed the accuracy of this baseline is low, but it has the lowest #Tokens. For instance, if we examine the #Tokens of Instruct ver. for GSM8K in Table 2, it shows approximately 3 times fewer tokens than DTO, even with on-par accuracy scores. I wonder if a simple in-context learning technique, such as providing few-shot demonstrations or Chain-of-Thought (CoT), would be more powerful than DTO in both accuracy and #Tokens. The authors should also analyze the result of #Tokens of Instruct ver. compared to DTO.

---

> ### Author Rebuttal · Authors · 2025-07-30
>
> **[W1 & Q2]**
> > **Reviewer’s Comment:** Please provide the justification for the choice of auxiliary model used in DTO.
>
> We thank the reviewer for this thoughtful and important comment. To clarify, our use of the auxiliary LLM (LLaMA 3.3-70B-Instruct) was intentionally aligned with the setup used in one of our primary baselines, **FCS + Ref.**, which also relies on the same model. By adopting the same auxiliary model, we ensure that the performance gains observed in our method are attributable to the proposed framework itself - rather than differences in model capacity - thereby enabling a fair and controlled comparison.
>
> We agree that the motivation for this design choice should be made more explicit. In the revised version, we will clarify this point to enhance the transparency and fairness of our experimental setup.
>
>
>
> **[W2]**
> > **Reviewer’s Comment:** Please explain the rationale behind segmenting reasoning trajectories using discourse cues that signal transitions in thinking, and possible strategies for handling cases where such cues are absent.
>
> Thank you for highlighting this important point. Firstly, we follow prior work [1, 2] in segmenting the reasoning trajectory into multiple thinking patterns using discourse cues such as *“Wait”* or *“Alternatively”*, which frequently appear in the outputs of large reasoning models [1, 2]. These cues are widely known to reflect shifts in the model’s reasoning process, including transitions into self-verification, intermediate summarization, or revised hypotheses as described in Section 3.
>
> We acknowledge, however, that this heuristic may not be applicable in all cases, particularly when such cue words are absent. In those situations, a promising extension would be to first analyze the lexical or structural patterns that commonly appear at the beginning of new paragraphs or sentences, and then leverage these patterns to define segment boundaries in a more systematic manner.
>
> While our current approach provides a simple and effective segmentation method grounded in observed discourse patterns, we agree that developing a more general and robust segmentation strategy would be a valuable direction for future work. We will clarify both this limitation and potential extensions in the revised version.
>
> **[W3]**
> > **Reviewer’s Comment:** Compare DTO with a simple in-context learning technique, such as providing few-shot demonstrations or CoTs to Instruct version.
>
> Thank you for the thoughtful comment. While we appreciate the interest in comparing DTO against instruction-tuned variants, we would like to firstly clarify the intended evaluation scope of our work. As described in Section 3.1, our primary goal is to improve the efficiency of the base Large Reasoning Model, e.g., DeepSeek-R1-Distill-Qwen-1.5B, while maintaining or improving its reasoning performance. Accordingly, the efficiency metric defined in Equation (14) is designed to measure gains relative to this shared base model.
>
> In contrast, the Instruct version is not directly derived from our target LRM but is instead based on its backbone model with separate instruction tuning. While we include Instruct ver. for completeness and to offer a broader perspective, it is not the main comparison target for efficiency evaluation, and we therefore do not compute the efficiency metric for them.
>
> To further assess the effectiveness of instruction-tuned models and the feasibility of simple prompting-based enhancements, we conducted additional experiments based on the reviewer’s suggestion. Specifically, using DeepSeek-R1-Distill-Qwen-1.5B as the base model, we evaluated its instruction-tuned version, Qwen2.5-Math-1.5B-Instruct, under two prompting strategies, using 10 examples sampled from the training set: **(1) few-shot demonstrations** and **(2) few-shot CoTs**.
>
> The results are presented in the table below. As shown, these prompting strategies do not lead to meaningful improvements in accuracy, and in some cases even degrade performance. This suggests that simply appending demonstrations or CoT examples may be insufficient for eliciting effective reasoning behavior in instruction-tuned models..
>
> We also observe that the Instruct variants tend to perform relatively well on simpler datasets such as GSM8K, where they achieve reasonable accuracy with strong efficiency. However, their performance drops significantly on more challenging benchmarks such as AMC and AIME, suggesting a lack of robustness in handling complex reasoning tasks. These trends imply that while instruction-tuned models can be effective for relatively straightforward tasks, their generalizability may be limited in settings that demand deeper reasoning. Investigating which types of models or prompting strategies best suit different task complexities could be an interesting direction for future work.
>
> We will include these additional results and analyses in the revised version to strengthen the comparison and highlight the advantages of our method.
>
> | Dataset |  | Instruct ver. + few-shot demos | Instruct ver. + few-shot CoTs | DTO (Ours) |
> | --- | --- | --- | --- | --- |
> | **MATH** | ACC | 73.91 | 76.47 | **85.48** |
> |  | #Tokens | 570.94 | 546.32 | 1936.19 |
> | **GSM8K** | ACC | 82.60 | **83.95** | 83.91 |
> |  | #Tokens | 318.37 | 307.04 | 844.18 |
> | **Gaokao** | ACC | 59.48 | 61.36 | **72.66** |
> |  | #Tokens | 589.41 | 590.32 | 2137.59 |
> | **AMC2023** | ACC | 48.12 | 50.00 | **70.25** |
> |  | #Tokens | 796.97 | 770.47 | 3376.98 |
> | **AIME2024** | ACC | 12.50 | 10.83 | **28.00** |
> |  | #Tokens | 1055.50 | 945.04 | 5877.44 |
> | **AIME2025** | ACC | 5.83 | 6.67 | **21.11** |
> |  | #Tokens | 896.23 | 816.89 | 5689.38 |
>
> **[Q1]**
> > **Reviewer’s Comment**: Can DTO handle queries where no golden label is available? I wonder whether DTO is applicable in such settings.
>
> Thank you for this insightful question. As described in the main paper, our current implementation of **DTO** leverages access to a ground-truth answer to supervise two stages:
>
> 1. **Identifying the appropriate point** at which the reasoning should be finalized, and
> 2. **Pruning intermediate thinking patterns** that do not meaningfully contribute to the reasoning objective.
>
> While our method currently operates in a supervised setting, we agree that it is meaningful to consider how our framework could be applied in weakly supervised or unsupervised settings. Importantly, **DTO can be naturally extended** to scenarios where ground-truth answers are unavailable, by utilizing a **reward model** as a surrogate for correctness. Below, we outline how each stage of DTO can be adapted accordingly.
>
> **1. Identifying the Finalization Point via Reward Comparison**
>
> In the absence of a golden answer, we can replace the answer-likelihood signal, which is defined in Equation (6) of the main paper, with a scalar reward function $R(\cdot)$, obtained via a reward model. For each truncated trajectory $\tau_i$ (as defined in Equation (4)), we compute its reward: $R(\tau_i)$.
> We also compute the reward for the full reasoning trajectory $y$, denoted $R(y)$.
> We then identify the earliest index $i'$ such that: $|R(\tau_i) - R(y)| \leq \epsilon$,
>
> where $\epsilon$ is a small threshold. This condition ensures that the partial trajectory is nearly as valuable (in terms of reward) as the full trajectory.
>
> We then truncate the reasoning at index $i'$ using the same binary selection function $f(\cdot)$ from Equation (7):
> $f(\delta_i) = 1$ if $i \leq i'$, $0$ otherwise.
> This enables DTO to finalize reasoning at an appropriate point based on reward saturation, rather than answer correctness.
>
> **2. Pruning Redundant Chunks via Reward Sensitivity**
>
> To adapt the pruning stage, we redefine the binary pruning function $g(\delta_i)$ using the reward sensitivity of each chunk:
> $g(\delta_i) = 1$ if $|R(\Delta^f_x \setminus \{\delta_i\}) - R(\Delta^f_x)| \leq \epsilon$, $0$ otherwise.
>
> That is, we remove a chunk $\delta_i$ only if its removal does not significantly degrade the reward. This allows us to safely discard segments that do not meaningfully contribute to high-reward outputs, while preserving critical content.
>
> Before applying this function, we may optionally employ the same auxiliary model used in the supervised setup to narrow down candidate chunks, identifying those likely to be redundant and reducing the risk of over-pruning.
>
> This reward-based extension maintains DTO’s core motivation while removing its reliance on ground-truth answers. It broadens the method’s applicability to **weakly supervised or unsupervised settings**, where annotated labels are unavailable.
>
> We appreciate the reviewer’s suggestion and will include a discussion of this extension in the revised version, as we believe it represents a promising direction for future work.
>
>
> ### References
>
> [1] Marjanović, Sara Vera, et al. "DeepSeek-R1 Thoughtology: Let's think about LLM Reasoning." *arXiv preprint arXiv:2504.07128* (2025).
>
> [2] Lu, Ximing, et al. "Retro-search: Exploring untaken paths for deeper and efficient reasoning." *arXiv preprint arXiv:2504.04383* (2025).

---

> > ### Comment · Reviewer_Nq5r · 2025-08-05
> >
> > I first appreciate the authors' response.
> >
> > **[W1 & Q2]**
> >
> > Thank you for clarifying that you selected the 70 B auxiliary LLM to match the FCS + Ref baseline. I understand the authors' choice. I still worry, however, that such a large model might amplify DTO’s gains in ways that smaller models cannot replicate. To isolate the contribution of your pruning strategy itself, could you also report results where the auxiliary LLM remains at the 1.5 B scale (e.g., DeepSeek-R1-Distill-Qwen-1.5B, DeepScaleR-1.5B-Preview, or another comparably sized model)? Such results would help readers judge how much improvement comes from the DTO framework versus the raw capacity of a larger assistant model.
> >
> >
> > **[W3]**
> >
> > Thank you for adding the few-shot ICL baselines. My original concern was that a simple in-context learning (ICL) strategy might surpass DTO in both 1) accuracy and 2) token efficiency in some cases. Your new results show that while the ICL variants are indeed more economical in tokens, DTO generally delivers higher accuracy. This addresses my concern.

---

### Official Review · Reviewer_iNNv · 2025-07-03

**Clarity:** 2
**Significance:** 2
**Originality:** 2
**Rating:** 3
**Confidence:** 4

**Summary:**

The paper present a novel method to improve quality of the reasoning in a dynamic manner. In particular, after model generating examples in the RL procedure, it refine these examples by cutting in the middle to only preserve the first-half part that are believed to be sufficient to derive the final answer. Additionally, it also presents another method to remove unnecessary reasoning steps in the middle, by testing if the correct answer can be obtained by removing the middle steps. Experimental evaluation shows that the proposed method effectively reducing the reasoning length while improving the reasoning accuracy as well.

**Questions:**

Removing in-the-middle reasoning steps:

Consider this reasoning pattern: A->B->C->D->Result. The D may contain sufficient information to obtain Results (many LLMs explicitly or implicitly generate results in D before finally conclude the results). But the fact is the B is necessary to obtain C and then D. Removing B is not OK and can ruin the model. So I think the method proposed in the paper is faulty.

**Ethical Concerns:**

["NO or VERY MINOR ethics concerns only"]

**Limitations:**

yes

**Quality:**

2

**Strengths And Weaknesses:**

The idea of refining the reasoning patterns seems reasonable to me, but the method to achieve this goal is not quite convincing. The model eventually learns from generated reasoning patterns and as a result improving the patterns is useful. In terms of the method proposed, I think the method of removing in-the-middle reasoning steps is dangerous to me.

---

> ### Author Rebuttal · Authors · 2025-07-30
>
> **[W1 & Q1]**
> > **Reviewer’s Comment**: Stage 2 of DTO might have a risk of over-pruning.
>
> We appreciate the concern regarding the risk of over-pruning. As described in Section 3, our method, DTO, consists of two stages:
>
> 1. Identifying the appropriate point at which the reasoning should be finalized, and
> 2. Pruning intermediate thinking patterns that do not meaningfully contribute to the reasoning objective.
>
> Stage 2, in particular, is **carefully designed to prevent harmful pruning** through a **conservative, multi-step verification process**. To begin, we assess the utility of each intermediate reasoning chunk using an auxiliary LLM. To avoid inadvertently removing valuable reasoning steps, we guide this LLM using explicitly conservative filtering criteria. As detailed in Appendix A.6, the prompt includes the original problem, the complete reasoning trajectory, the correct final answer, and the following instructions:
>
> - **Retain** any chunk that contributes meaningfully, including narrowing down possibilities, offering partial insight (even if incomplete), performing sub-computations, supporting downstream reasoning, or even reinforcing understanding through redundancy.
> - **Remove** only those chunks that are clearly off-topic, factually incorrect in a harmful way, or provide no meaningful content - with the default bias *toward retaining* unless strong justification for removal is found.
>
> To further verify the safety of each removal, we perform a lightweight decoding-based validation. For each candidate chunk $\delta_i$ - previously identified by the auxiliary model as a removable thinking pattern - we define the following sequence:
> $\tilde{\Delta}_x^{f\setminus \delta_i} = \left[\ \delta_j \in \tilde{\Delta}_x^f \middle| \delta_j \ne \delta_i,\ \text{and before ``}\backslash\text{boxed''} \ \right]$
>
> We then quickly decode from this sequence to check whether the model can still produce the correct answer. The pruning of $\delta_i$ is accepted only if the answer remains correct; otherwise, the chunk is retained.
>
> As shown in Tables 1 and 2 of the main paper, this careful procedure enables **DTO to consistently outperform baselines in terms of reasoning efficiency**, achieving similar or better accuracy while generating fewer tokens.
>
> Additionally, motivated by your comment, we conducted an **ablation study** to assess the isolated impact of Stage 2, using DeepSeek-R1-Distill-Qwen-1.5B as the base model. Below, we present the results comparing DTO with only Stage 1 to the full version with both stages applied:
>
> | Dataset |  | Base | DTO (Stage 1) | DTO (Stage 1 + 2) |
> | --- | --- | --- | --- | --- |
> | **MATH** | ACC | 79.8 | 85.21 | 85.48 |
> |  | #Tokens | 3543.44 | 2256.26 | 1936.19 |
> |  | Eff. | 1.000 | 1.677 | **1.960** |
> | **GSM8K** | ACC | 82.13 | 83.95 | 83.91 |
> |  | #Tokens | 1382.99 | 1050.54 | 844.18 |
> |  | Eff. | 1.000 | 1.346 | **1.674** |
> | **Gaokao** | ACC | 66.62 | 72.79 | 72.66 |
> |  | #Tokens | 3725.16 | 2559.51 | 2137.59 |
> |  | Eff. | 1.000 | 1.590 | **1.901** |
> | **AMC2023** | ACC | 58.25 | 70.00 | 70.25 |
> |  | #Tokens | 5338.54 | 3768.81 | 3376.98 |
> |  | Eff. | 1.000 | 1.702 | **1.907** |
> | **AIME2024** | ACC | 21.44 | 30.00 | 28.00 |
> |  | #Tokens | 7359.24 | 6301.40 | 5877.44 |
> |  | Eff. | 1.000 | 1.634 | **1.635** |
> | **AIME2025** | ACC | 18.89 | 21.67 | 21.11 |
> |  | #Tokens | 7236.66 | 6237.63 | 5689.38 |
> |  | Eff. | 1.000 | 1.331 | **1.421** |
>
> These results show that **adding Stage 2 on top of Stage 1 has minimal impact on accuracy, while improving efficiency**. This confirms that our pruning procedure removes redundant or distracting reasoning without harming the underlying logic.
>
> We appreciate the reviewer’s thoughtful comment again, which prompted us to conduct this additional analysis. We will include these results and clarifications in the revised version to more clearly communicate the safety and effectiveness of our pruning strategy.

---

> ### Author Response · Authors · 2025-08-06
>
> Dear Reviewer,
>
> We sincerely appreciate your thoughtful feedback and the time you've devoted to our paper. As the discussion phase draws to a close, we would like to check if our response has sufficiently addressed your comments. If there are any remaining concerns or further suggestions you'd like us to consider, we would be grateful to hear them.
>
> Your insights have been extremely helpful in improving our work, and we're committed to refining the paper as best we can.
>
> Thank you again for your time and contributions.
>
> Best,
>
> Authors

---

### Comment · Area_Chair_cPRi · 2025-08-04
**Gentle Reminder: Please Reply to Authors’ Responses (Only if Not Yet Done)**

Dear Reviewers,

As the discussion deadline approaches, may we kindly ask you to review the authors’ responses and post a constructive reply—unless you have already done so, in which case please kindly disregard this gentle reminder.

Your thoughtful engagement is deeply appreciated and essential to a fair and timely process. With sincere thanks for your continued dedication.

Area Chair

---

### Decision · Program_Chairs · 2025-09-17

**Decision:**

Accept (poster)

**Comment:**

Summary of the paper: This paper introduces Dynamic Thinking Pattern Optimization (DTO), a framework that treats long reasoning traces as sequences of modular “thinking patterns.”  Using Monte-Carlo sampling, DTO first identifies the earliest prefix of a trace that still yields the correct answer, truncating the remainder.  It then feeds the remaining trace into an auxiliary LLM that prunes any intermediate steps whose removal does not harm correctness.  The resulting minimal traces are used to build a preference dataset that is optimized via RL.  Across mathematical benchmarks and two 1.5 B-parameter LRMs, DTO reduces attention FLOPs by up to 47 % and token counts from ~5,000 to ~3,000 while maintaining accuracy on originally-correct problems and improving accuracy by 15.6 % on originally-incorrect ones; preference optimization further yields up to 12 % absolute accuracy gains.

Strengths of the paper:
1. Timely problem framing: directly tackles the emerging over-thinking issue in LRMs by formulating reasoning-path compression as a constrained optimization problem. The proposed two complementary mechanisms—early-stopping via Monte-Carlo estimation and LLM-based segment pruning—are simple, model-agnostic, and empirically effective. The preference-optimization pipeline converts the distilled traces into high-quality supervision, opening a path for continual self-improvement.
2. Strong empirical results: simultaneous gains in efficiency (FLOPs/tokens) and accuracy, including the valuable ability to repair previously incorrect chains. Rich ablations and statistical analyses lend credibility to the findings.
3. Presentation: The paper is clearly written and content-rich, making the method easy to reproduce.

Weaknesses of the paper: After reading the rebuttal, I think the authors have addressed the major concerns raised by all the reviewers. Except for Reviewer iNNv, he did not reply or even acknowledge the rebuttal. Please incorporate the content during rebuttal into the camera-ready version of the paper.

Reasons for the decision: This work presents a clear, principled method for shortening and refining lengthy reasoning traces while simultaneously improving accuracy—a key advance toward more efficient and reliable LRMs. Strong experimental results, a reproducible design, and broad potential offset the noted limitations, and all three responsible reviewers expressed clear support after the rebuttal and area-chair discussion. I therefore lean towards accepting this paper.